# Artificial intelligence guided discovery of a barrier-protective therapy in inflammatory bowel disease

Debashis Sahoo [1,2,3,10✉], Lee Swanson[4,10], Ibrahim M. Sayed[5,9,10], Gajanan D. Katkar[4], Stella-Rita Ibeawuchi[6], Yash Mittal[7], Rama F. Pranadinata[4], Courtney Tindle[4], Mackenzie Fuller[4], Dominik L. Stec[4], John T. Chang [7], William J. Sandborn[7], Soumita Das [6✉] & Pradipta Ghosh [3,4,7,8✉]

Modeling human diseases as networks simplify complex multi-cellular processes, helps understand patterns in noisy data that humans cannot find, and thereby improves precision in prediction. Using Inflammatory Bowel Disease (IBD) as an example, here we outline an unbiased AI-assisted approach for target identification and validation. A network was built in which clusters of genes are connected by directed edges that highlight asymmetric Boolean relationships. Using machine-learning, a path of continuum states was pinpointed, which most effectively predicted disease outcome. This path was enriched in gene-clusters that maintain the integrity of the gut epithelial barrier. We exploit this insight to prioritize one target, choose appropriate pre-clinical murine models for target validation and design patient-derived organoid models. Potential for treatment efficacy is confirmed in patient-derived organoids using multivariate analyses. This AI-assisted approach identifies a first-in-class gut barrier-protective agent in IBD and predicted Phase-III success of candidate agents.

[1] Department of Pediatrics, University of California San Diego, San Diego, CA, USA. [2] Department of Computer Science and Engineering, Jacob's School of Engineering, University of California San Diego, San Diego, CA, USA. [3] Rebecca and John Moore Comprehensive Cancer Center, University of California San Diego, San Diego, CA, USA. [4] Department of Cellular and Molecular Medicine, University of California San Diego, San Diego, CA, USA. [5] Department of Medical Microbiology and Immunology, Faculty of Medicine, Assiut University, Assiu, Egypt. [6] Department of Pathology, University of California San Diego, San Diego, CA, USA. [7] Department of Medicine, University of California San Diego, San Diego, CA, USA. [8] Veterans Affairs Medical Center, La Jolla, CA, USA. [9] Present address: Department of Pathology, University of California San Diego, San Diego, CA, USA. [10] These authors contributed equally: Debashis Sahoo, Lee Swanson, Ibrahim M. Sayed. ✉email: dsahoo@ucsd.edu; sodas@ucsd.edu; prghosh@ucsd.edu

D rug discovery, in its current state, is wasteful and fraught with increasing trends of failures, implying the existence of uncertainty and impreciseness in the process[1]. The use of computational algorithms to sort through 'big data' (such as transcriptomics) and build networks to visualize the complexity has been a popular approach to understand complex human diseases and even prioritize targets. First, relationships are identified between pairs of genes using symmetric computational frameworks such as linear regression, dimension reduction, and clustering. Subsequently, gene co-expression networks (GCNs) are built by focusing on pairwise gene similarity scores that meet a set statistical threshold. GCN-based analyses severely influenced by the above techniques for connecting two nodes with an edge[2–12] have helped formalize Network Medicine as a field[13,14] and deliver many successes [in drug repositioning, drug-target discovery, drug-drug interactions, side effect predictions, etc.; reviewed in[15]]. Identification of drugs that can predictably re-set the network in complex multi-component diseases has been a topic of intense investigation for decades, resulting in novel targets[16–20].

Inflammatory bowel disease (IBD) is an example of a complex, multi-factorial, chronic condition with urgent and unmet needs, where network-based analyses can have impact. IBD is an autoimmune disorder of the gut in which diverse components (microbes, genetics, environment and immune response) intersect in elusive ways and culminate in overt disease[21]. It is also heterogeneous with complex sub-disease phenotypes (i.e., strictures, fistula, abscesses, and colitis-associated cancers). Currently, patients are offered inflammation-reducing therapies that have a widely variable ~18–58% response-rate [where the response can be either clinical or endoscopic, with a placebo rate of 3–30% during induction therapy[22]]; 40% of responders become refractory to treatment within one year[23]. None of the current therapies focus on the most widely recognized indicator/predictor of disease relapse, response and remission[24–29], i.e., a compromised epithelial barrier.

Here we present a different network-based approach for drug discovery that uses artificial intelligence (AI) to prioritize target identification and then guides its subsequent validation in network-rationalized preclinical mouse and patient-derived organoid models in 4 steps (Fig. 1). We demonstrate how these four steps synergize and aid in the modeling of fundamental progressive time series events underlying complex human diseases and exploit such insights to improve precision when developing disease-modifying drugs.

## Results and discussion

**A Boolean Network for IBD reveals epithelial barrier disruption as an invariant continuum event in IBD.** Using publicly available IBD-tissue derived transcriptomic datasets, representing heterogeneous samples (Supplementary Data 1), a Boolean Implication network is built (see Methods; and Supplementary Text). As expected, the IBD-Boolean implication network (IBD-map; Fig. 2, Supplementary Fig. S1A, B; Supplementary Data 2) showed scale-free architecture (Supplementary Fig. S1C), i.e., there are few large clusters, whereas the majority are smaller sized clusters. *BoNE*-enabled exploration of the Boolean paths (Supplementary Method; Fig. 2B; Supplementary Fig. S1D–F) revealed how some of the biggest clusters are connected by a series of BIRs (Green-Red arrows/Black-Blue lines, Fig. 2C). Reactome pathway analysis of these clusters along the path continuum revealed the most important biological processes with which they associate (Fig. 2C; Supplementary Data 2). Each cluster was then evaluated for whether they belong to the healthy or diseased side depending on whether the average gene expression value of a cluster in

heathy samples is up or down, respectively. The clusters were then arranged sequentially from healthy on the left side to disease on the right side, allowing for the modeling and visualization of a time-series of biological processes during the initiation and progression of disease, similar to previously published models of B cell differentiation using Boolean Implication Networks, *BIN*, built using if-then relationships[30]. This effort yielded a map of IBD (Fig. 2C). A time series of IBD-associated invariant events emerged— epithelial tight junctions (TJs) and other types of cell-cell junctions appeared leftmost on the healthy side (C#1–2) of the IBD-map (Supplementary Data 2), levels of which are downregulated early during disease initiation and are progressively lost. This is followed by bioenergetic stress (C#3), culminating in inflammation and fibrosis mediated via the activation of both innate and adaptive immune components and pathways that lead to the formation, resorption and control cellular response to the extracellular matrix (ECM) (C#4–6) (Fig. 2C).

**Machine learning identified epithelial barrier-related gene clusters as predictors of therapeutic response.** Next, we introduced in *BoNE* machine learning that seeks to identify which of the gene clusters (nodes) connected by Boolean implication relationships (edges) are most optimal in distinguishing healthy from diseased samples. *BoNE* computes a score that naturally orders the samples; this score can be thought of as a continuum of states. Among all possible permutations and combinations, clusters #1-2-3 (C#1-3) emerged as the best in separating normal healthy from IBD-afflicted samples (Fig. 2D, Supplementary Fig. S1G) with the highest accuracy (Fig. 2E). As expected of the invariant nature of the Boolean relationships, this C#1-3 signature performed consistently well across all the independent training ($n = 3$) and validation ($n = 6$) cohorts (Supplementary Fig. S2A, B). We compared our approach directly with differential and Bayesian approaches; the latter was optimized by Peters et al.[17] for the analysis of IBD datasets. Despite minimal overlaps between differentially regulated genes across these independent cohorts (Supplementary Fig. S3), conventional approaches e.g., differential and Bayesian performed equally well in separating the heathy and IBD-afflicted samples (Supplementary Fig. S4A, B). However, when it came to distinguishing responders from non-responders in the only prospective study to date where colons were analyzed by RNA-Seq *prior* to the initiation of treatment with TNFα-neutralizing mAbs [E-MTAB-7604[31]], Boolean analysis was more accurate than the other two approaches [Fig. 2F; ROC-AUC for Boolean, 0.91; Differential, 0.73; Bayesian, 0.64]. These findings indicate that the Boolean approach may be superior in predicting therapeutic response [response was defined as endoscopic remission[31]]. Additionally, *BoNE* revealed the ability of the C#1-3 signature to segregate samples according to the aggressiveness of disease consistently across five additional validation cohorts (Fig. 2G); it could separate active from inactive disease[32,33], responders from non-responders receiving two different biologics, Infliximab[34] or Vedolizumab[35], and even distinguished those with quiescent disease with or without remote neoplasia[36] (Fig. 2G). These findings demonstrate the power of Boolean networks in accurately modeling gene expression changes that occur during IBD pathogenesis and predicting clinical outcomes.

*Network-rationalized selection of PRKAB1 as a barrier-protective target in IBD.* Next, we sought to exploit the predictive power of *BoNE* for rationalized target identification and drug discovery. The IBD-map (Fig. 2C) and multiple validation studies (Fig. 2F–G) concur that healthy controls and diseased patients in remission share a common signature– high expression of genes in

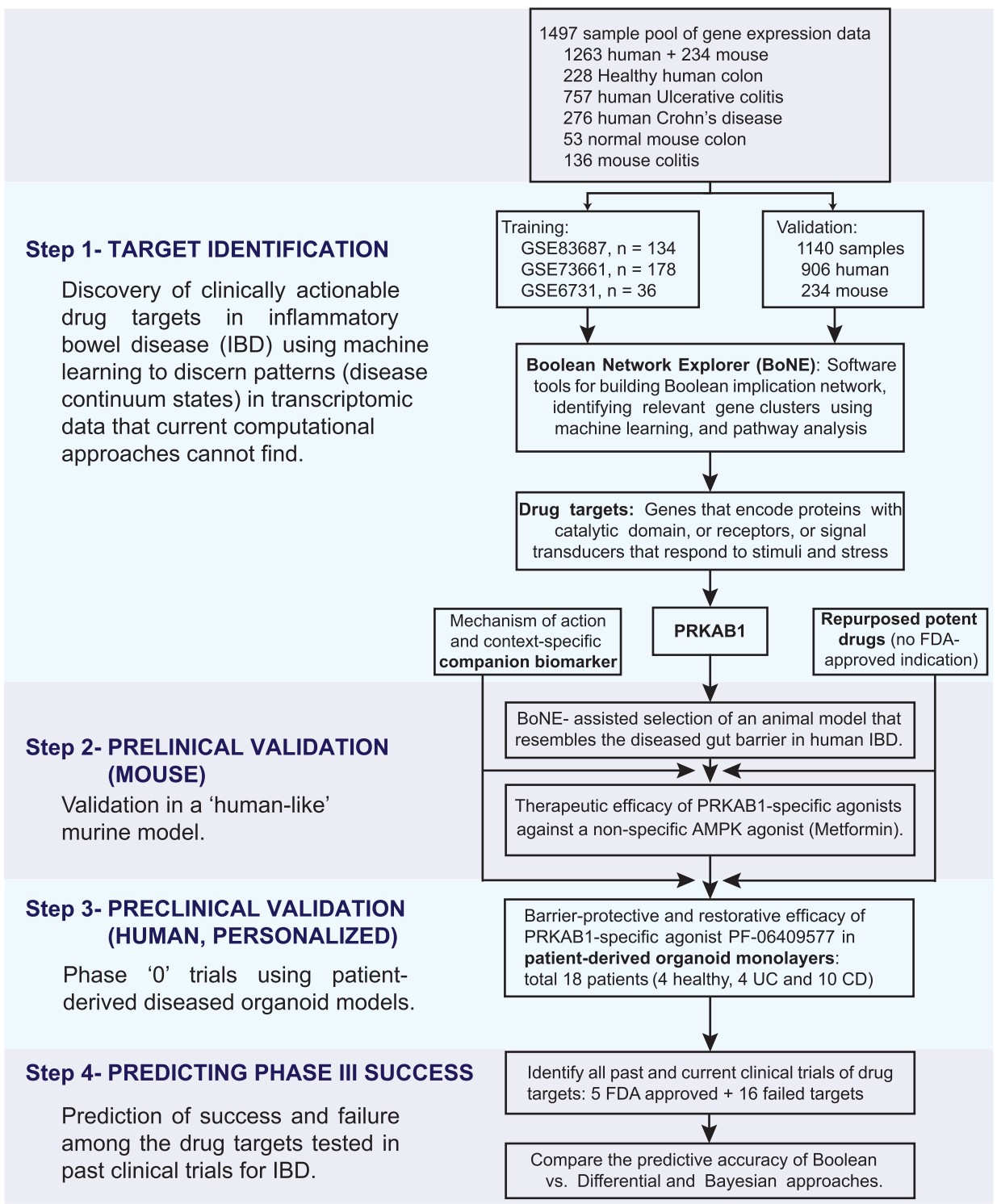

**Fig. 1 Study design.** A database containing 1497 human gene–expression data from both 1263 human samples and 234 mouse samples was mined to build a validated Boolean implication network-based computational model of disease continuum states in inflammatory bowel disease (IBD) [Supplementary Data 1 lists all the datasets, showcasing sample heterogeneity]. Paths, clusters and a list of genes in the network-based model are prioritized to discover clinically actionable drug targets. The search yield PRKAB1 as a barrier-protective therapeutic target. Two PRKAB1-specific agonists were successfully tried in mice. In phase '0' trials, one of the PRKAB1-agonist PF-06409577 was successfully used in patient-derived organoid models. Furthermore, the network-based model accurately classified the FDA-approved vs. the failed targets.

C#1-3 and low expression of genes in C#4-6, whereas patients with active disease show the opposite pattern. Because the Boolean implication relationships between C#1-3 and C#4-6 are 'opposite', pharmacologic activation of gene products from C#1-2-3 is predicted to both promote C#1-2-3 (healthy) and inhibit

C#4-6 (disease) gene signatures. We next prioritized target genes in C#1-2-3 based on 4 commonly used methods: i) 'druggability' of the targets; ii) availability of potent and specific compounds; iii) sound contextualized biological rationale; and vi) availability of companion markers. To assess 'druggability', gene ontology

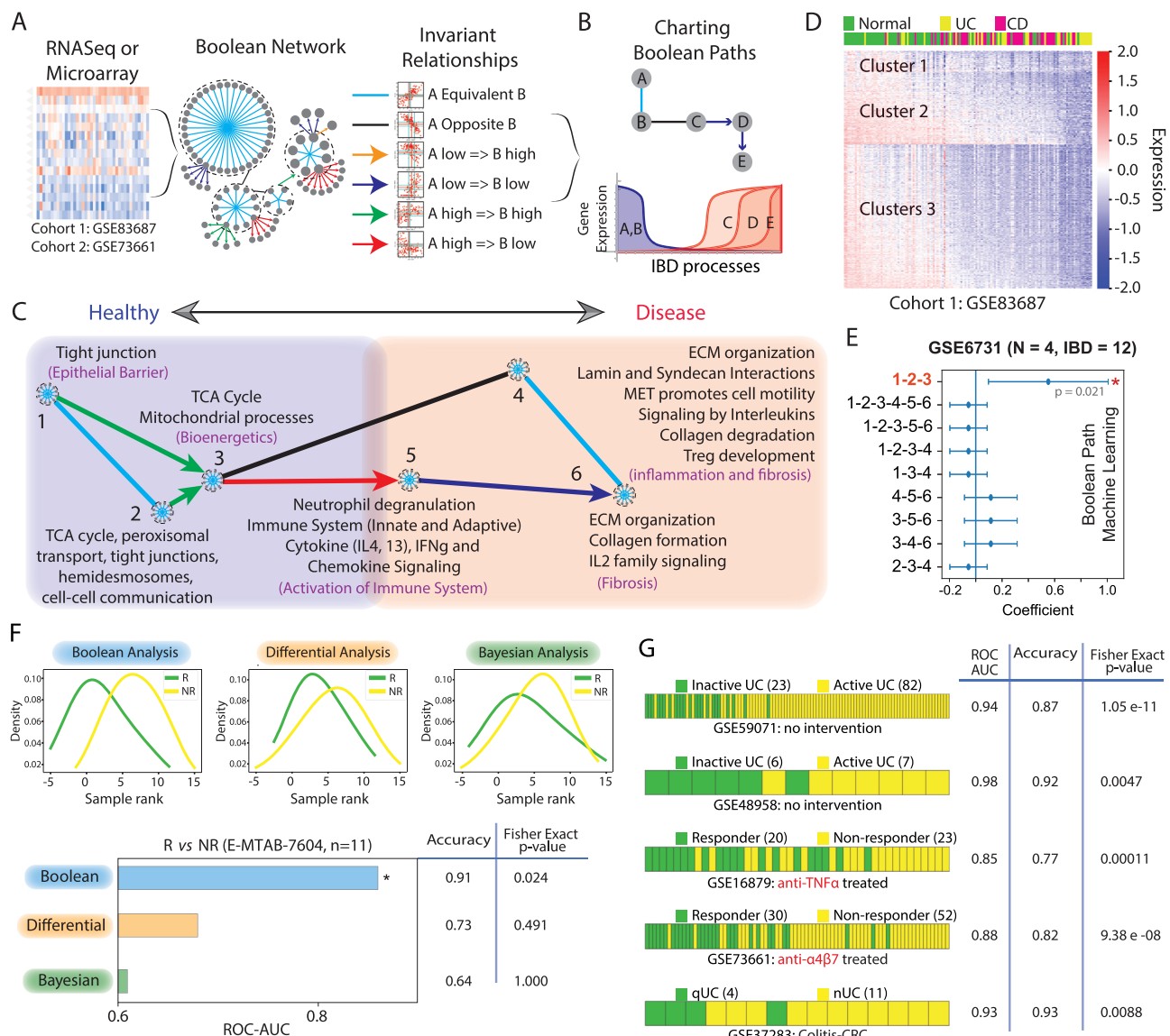

**Fig. 2 Generation and validation of Boolean Network map of IBD. A** We applied Boolean Network Explorer (*BoNE*; see also Supplementary Fig. S1) to analyze two IBD datasets: GSE83687 and GSE73661 [17,35]. The Boolean network (middle) contains the six possible Boolean relationships (right) in the form of a directed graph with genes having similar expression profiles organized into clusters and relationships between gene clusters represented as color-coded edges. **B** Schematic illustrating how Boolean cluster relationships are used to chart disease paths (Top); and individual gene expression changes along a boolean path, illustrating gene expression dynamics within the normal to IBD continuum (bottom). **C** Reactome pathway analysis of each cluster along the top continuum paths was performed to identify the signaling pathways and cellular processes that are enriched during IBD progression. [Supplementary Data 2 lists the genes and pathways associated with each gene cluster]. **D** Heatmap of the expression profile of genes in GSE83687 using Boolean clusters (C#1-2-3) superimposed on sample type (top bar) demonstrates the accuracy of Boolean analysis in sample segregation into normal and IBD. **E** Selection of Boolean path using machine learning. Linear regression on Test dataset 3 (GSE6731) was used to select the best path that can separate normal and IBD samples. Coefficient of each path score (at the center) with 95% confidence intervals (as error bars) and the *p* values were illustrated in the bar plot. The *p* value for each term tests the null hypothesis that the coefficient is equal to zero (no effect). See also Supplementary Fig. S2 for the performance of the selected Boolean path, Clusters #1-2-3, on numerous independent test and validation cohorts. **F** Direct comparison of Boolean, Differential and Bayesian analysis in predicting responders (R) and non-responders (NR) to anti-TNFα treatment in E-MTAB-7604 (*n* = 11) dataset. **G** Prediction of active *vs.* inactive UC (GSE59071, GSE48958), responder *vs.* non-responder to treatment with anti-TNFα (GSE16879) and anti-α4β7 (GSE73661) mAbs, quiescent UC without or with remote neoplasia (GSE37283, *N* = 20) by using Boolean analysis. Fisher exact test (two-sided) is performed on a 2 × 2 contingency table based on the prediction. Source data are provided as a Source Data file.

(GO) molecular function analysis of C#1-3 was carried out, identifying receptors, enzymes and signal transducers that can be targeted easier than other molecules (Fig. 3A). Of these druggable interfaces, 17 targets were identified as associated with GO biological function of 'response to stress'. Two of 17 were kinases, of which only one, PRKAB1(β1 subunit of the metabolic master regulator, AMPK) had commercially available and extensively

validated specific and potent agonists with known structural basis[37–39] (Fig. 3A). When proteins encoded by C#1-6 were analyzed for cooperativity between cellular processes within protein-protein interaction (PPI) networks using STRING[40], PRKAB1 and other subunits of AMPK appeared at the crossroads between 'pathogen-sensing', 'autophagy' and epithelial 'tight and adherens junctions' and 'polarity complexes', modules

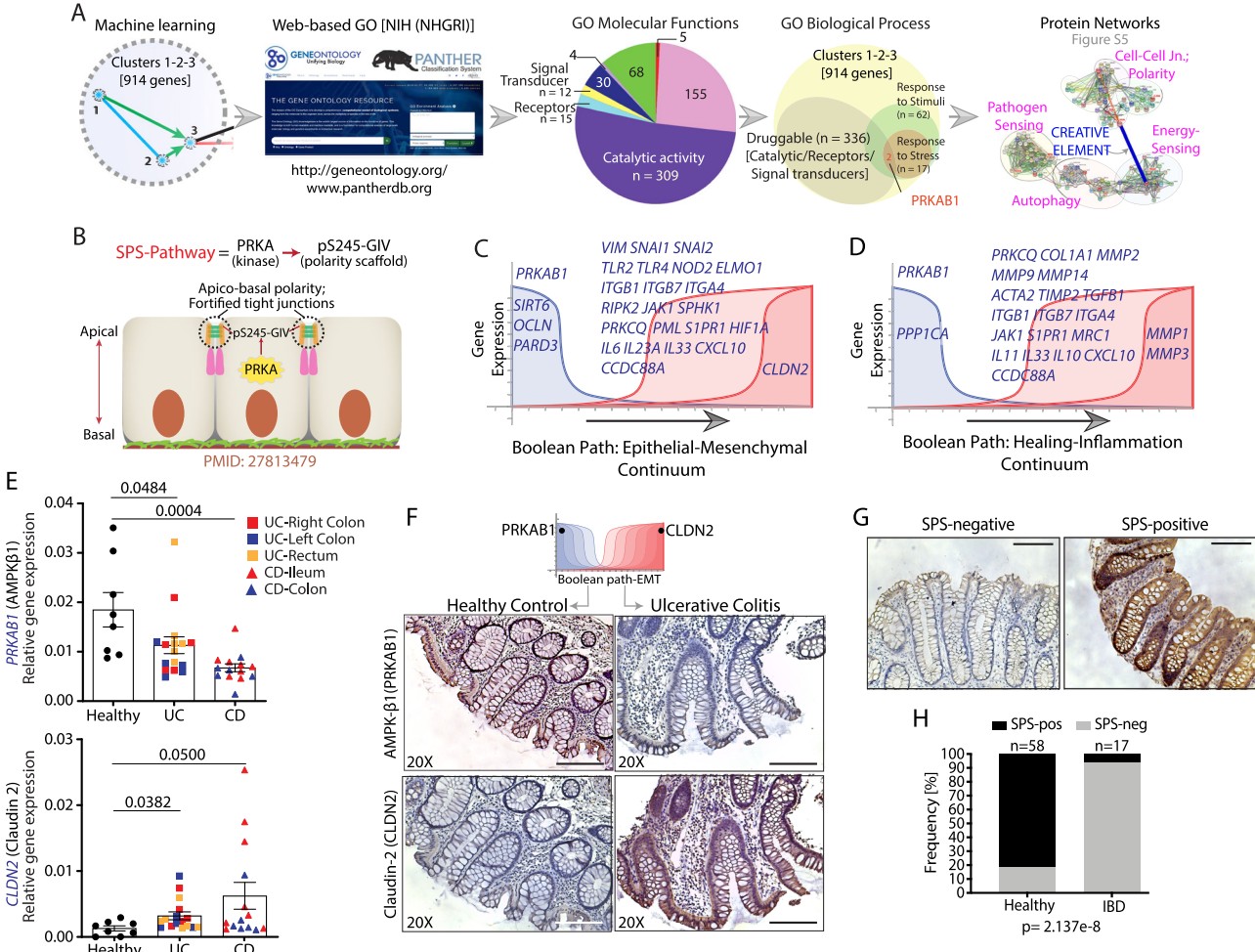

**Fig. 3 Identification of PRKAB1 as a therapeutic target to promote intestinal barrier function. A** Computational workflow for identification of PRKAB1 as a druggable target for promoting a barrier-protective transcriptional program within the IBD network. **B** Schematic summarizing key components of the SPS-pathway used in fortifying tight junctions and promoting apical-basal polarity. **C, D** Detailed view of two prominent disease paths identifying downregulation of PRKAB1 as a shared early event in IBD pathogenesis [see a complete list of genes in Table 1]. **E** The expression of PRKAB1 and CLDN2 transcript levels (qPCR) was assessed in the ileum and colon biopsies of IBD patients (UC = 16 and CD = 14) or healthy controls (n = 8). The different locations of the tissue specimens from UC and CD subjects were shown in different colors, as stated in the figure. Data represented as mean ± S.E.M. and two-tailed Mann–Whitney test using $p = 0.05$ cutoff was used to calculate significance. **F** IHC of IBD patient colon biopsies assessed for expression of PRKAB1 and CLDN2. Representative images are shown [expanded selection of IHC in Supplementary Fig. S9]. **G** Activation status of SPS-pathway was analyzed in FFPE colon biopsies from UC/CD patients using an anti-pS245-GIV antibody. Representative images are shown. Scale bars = 100 μm. The method for assessing the SPS-pathway and details regarding the phospho-specific antibody is included in Supplementary Fig. S10. **H** Bar graphs showing the frequency of SPS pathway activation in healthy vs. IBD patients. Two-sided Fisher's exact test was used to calculate significance. Source data are provided as a Source Data file.

(Supplementary Fig. S5). This was hardly surprising because AMPK's role in the stabilization of the gut barrier has been known for over a decade[41–44]; however, beyond weak agonists like mesalamine[44,45], the use of the other relatively more potent but non-specific agonists such as Metformin in IBD is limited due to symptoms of GI intolerance.

*Proposed mechanism of action of PRKAB1 in the gut lining.* We hypothesized that our AI-guided approach may have pinpointed a specific subtype of AMPK (i.e., trimers of the kinase that includes PRKAβ1) that is important and that PRKAB1-specific agonists may offer a higher degree of precision and efficacy over non-specific AMPK agonists. Mechanistically, they may augment epithelial tight junctions (TJs) in the presence of pathogens by activating a specialized signaling program in the epithelium lining the gut, the stress polarity signaling (SPS) pathway[46] (Fig. 3B). The SPS pathway involves the phosphorylation of the polarity

scaffold, *Girdin* (GRDN), at a single site (Ser[245]) by AMPK, an event that appears to be both necessary and sufficient for the strengthening of epithelial junctions under bioenergetic stress. Because the SPS-pathway is triggered exclusively as a stress response and improves modular cooperativity within the PPI network, it fulfills the criteria of "creative elements"[47]; the latter is believed to be critical for the evolvability of complex systems and their pharmacological modulation is predicted to help survive unprecedented challenges/stressors. More importantly, we recently confirmed using PRKAB1-specific agonists that the SPS-pathway serves as a putative cell-type-specific 'companion' biomarker for AMPK activation in the gut epithelium[48].

*Predicted impact of PRKAB1-agonists on the gut lining.* To determine how PRKAB1-agonists may impact the two progressive pathognomonic features of IBD: 1) Epithelial dysfunction and mesenchymal transition (EMT), which distinguishes active

from inactive lesions[49], and 2) inflammation and fibrosis, we explored disease continuum paths within the IBD-network by accessing another feature of *BoNE*. Given a set of genes in any process, *BoNE* can identify and help visualize how their levels of expression change along a linear path based on the Boolean implication relationships. The EMT-continuum (Fig. 3C; Table 1) showed suppression of key TJ/polarity genes (*OCLN, PARD3*) is permissive to the upregulation of pro-inflammatory cytokines (i.e., *IL6, IL23A, CXCL10*), inflammatory trafficking molecules (i.e., *ITGB1, ITGB7, ITGA4, S1PR1*), pathogen-sensing pathways (i.e., *TLR2/4, NOD2, ELMO1*), and EMT genes (i.e., *VIM, SNAI1/2*), culminating in leakiness of the barrier, as evidenced by increase in the pore-forming leaky tetraspanin, *CLDN2*. The healing-inflammation continuum (Fig. 3D; Table 1) showed loss of C#1-2 genes (*PRKAB1, PPP1CA*) is permissive to proinflammatory signaling factors (i.e., *PRKCQ, JAK1, MRC1*), cytokines (i.e., *IL11, IL33, IL10*), inflammatory trafficking molecules (i.e., *ITGB1, ITGB7, ITGA4*), pro-fibrotic factors (i.e., *COL1A1, PRKCQ, ACTA2, TIMP2, TGFB1*), and matrix metalloproteinases (i.e., *MMP2, MMP9, MMP14, MMP1, MMP3*). *PRKAB1* was present in both disease paths; its activation was predicted to augment epithelial polarity and TJ integrity that are controlled by C#1-3, thereby restoring the integrity of the gut barrier and suppressing the two progressive pathophysiologic changes in IBD that are controlled by C#4-6, namely, EMT and inflammation/fibrosis. Although the algorithm tries to uncover a timeseries component of the IBD events, the algorithm is unable to pick the direction, start and end of these events; the network direction is oriented later by revealing the identity of the sample types that overwhelmingly cluster at one end *vs*. the other. Our analysis simply shows what is common knowledge in IBD, i.e., if the barrier is disrupted, then it can be permissive to inflammation; the reverse is also true that is if there is inflammation, that can lead to barrier disruption. Therefore, a logical interpretation of the Boolean paths is that the state of no inflammation and intact mucosa is both the start point of the disease and the desired end point of therapeutic goals. The algorithm, for the first time, precisely lists actionable genes/targets which may help achieve that goal; in this case, PRKAB1-agonists were predicted to work through upholding epithelial polarity and TJ integrity, which in turn should reduce inflammation.

*Expression pharmacology studies rationalize the use of PRKAB1-agonists in IBD.* We noted that an IBD-associated SNP has been reported for *PRKAB1*, but no other subunit of AMPK (Supplementary Fig. S6A, B). It was also the only subunit of AMPK that is downregulated in IBD (Supplementary Fig. S6C, D). Target transcript analysis by quantitative PCR (qPCR) from human colon biopsies showed a significant decrease in *PRKAB1* and a concomitant increase in *CLDN2* expression in IBD-afflicted tissues, representing both UC and CD, regardless of disease location (Fig. 3E). Analysis of two other independent cohorts also concurred, i.e., decreased expression of *PRKAB1* transcripts in IBD was associated with a concomitant increased expression of *CLDN2* in inflamed regions of the colon (Supplementary Fig. S6E). Furthermore, target expression analyses confirmed that low levels of *PRKAB1* correlate with a higher degree of leakiness of the epithelial barrier (*CLDN2*), proinflammatory cytokines (*MCP1, IL8, IL6* and *TNFα*) and higher expression of a mucosal gene signature that predicts non-response to anti-TNFα[50] (Supplementary Fig. S7).

Target protein expression analyses studies were performed via three approaches. First, we noted that unlike its counterpart, AMPKβ2, *PRKAB1*-encoded AMPKβ1 is preferentially expressed in the gut (and not liver and skeletal muscle, two major sites for the metabolic action of AMPK), as determined using two different antibodies [Human Protein Atlas (www.proteinatlas.org); Supplementary Fig. S8]. Second, our immunohistochemistry (IHC) studies on human colon biopsies revealed that compared to healthy controls, patients with IBD display decreased AMPKβ1 (PRKAB1) and increased claudin-2 (CLDN2) staining at the apical side of the epithelial barrier (Fig. 3F, Supplementary Fig. S9A). Third, analysis of a previously published proteomics dataset from IBD-afflicted patients[51] further confirmed that diseased colons have high or low expression levels of AMPKβ1 depending on disease activity (Supplementary Fig. S9B).

We next asked if the proposed epithelium-specific mechanism of action of PRKAB1-agonists, i.e., their ability to activate the SPS-pathway, is relevant in IBD. IHC on FFPE colon biopsies from healthy and IBD-afflicted patients using a previously validated antibody (Supplementary Fig. S10) revealed that the SPS-pathway is more frequently suppressed in IBD compared to healthy controls (Fig. 3G, H), suggesting that this barrier-protective pathway may be compromised during IBD pathogenesis. Together, these expression studies further rationalize the selective activation of PRKAB1 as a therapeutic strategy to enhance the gut barrier function in IBD.

*PRKAB1-agonists ameliorate colitis in a network-rationalized murine model.* It is well known that no single mouse model recapitulates *all* the multifaceted complexities of IBD[52]. AMPK's role (or the role of its agonists) in protecting the gut barrier has been evaluated in several murine models of colitis, including DSS[44], TNBS[53–55], IL10−/−[56,57] and adoptive T-cell transfer models[58]. We used *BoNE* to prioritize the murine models of colitis that most accurately recapitulates the barrier-defect transcript signature in human IBD, i.e., downregulation of genes in C#1-3 (Fig. 4A, B; Supplementary Data 1). DSS-induced colitis, which triggers intestinal inflammation by compromising the integrity of the gut barrier[59] emerged as the best (for both bulk colon and sorted epithelial cell-derived datasets), closely followed by TNBS, adoptive T-cell transfer and *Citrobacter*-induced colitis,

**Table 1 Genes that share a strong Boolean implication relationship with PRKAB1 on the major continuum paths in IBD (related to Fig. 3C, D).**

**1A. Epithelial Mesenchymal Transition (EMT) Continuum**

| CCDC88A | IL17B | OCLN | SIRT1 |
|---|---|---|---|
| CDH1 | IL23A | PARD3 | SIRT6 |
| CLDN2 | IL6 | PML | SMAD7 |
| CXCL10 | IL6 | PPARG | SNAI1 |
| ELMO1 | ITGA4 | PPARGC1A | SNAI2 |
| F11R | ITGB1 | PRKAB1 | SPHK1 |
| HIF1A | ITGB7 | PRKCQ | TLR2 |
| HMOX1 | JAK1 | RIPK2 | TLR4 |
| IL12A | MADCAM1 | S1PR1 | VIM |
| IL17A | NOD2 | S1PR5 | |

**1B. Healing - Fibrosis Continuum**

| PRKAB1 | MRC1 | MMP3 | S1PR1 |
|---|---|---|---|
| PPARG | IL33 | MMP9 | S1PR5 |
| PPP1CA | IL6 | MMP12 | JAK1 |
| PPARGC1A | CCL2 | MMP13 | SMAD7 |
| CCDC88A | TNF | MMP14 | IL17A |
| PRKCQ | COL1A1 | IL11 | IL17B |
| ACTA2 | TIMP1 | ITGA4 | IL12A |
| TGFB1 | TIMP2 | ITGB1 | IL23A |
| IL10 | MMP1 | ITGB7 | IL6 |
| IL1B | MMP2 | MADCAM1 | RIPK2 |

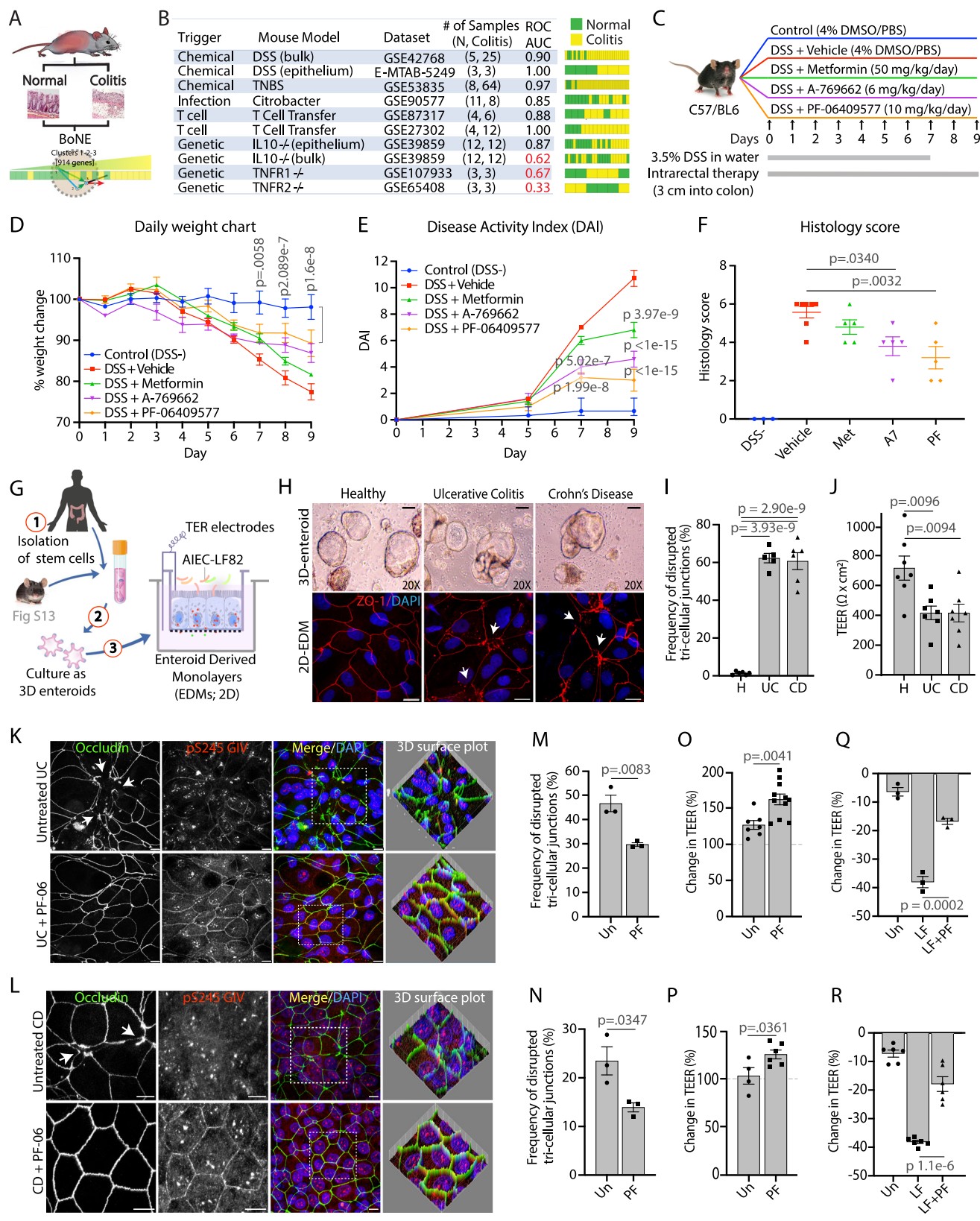

whereas the genetic models were deemed inferior (Fig. 4B). To test if PRKAB1-agonists can protect the barrier against stress-induced collapse, mice were treated intra-rectally with DMSO alone (vehicle control), metformin (non-specific AMPK agonist), or PRKAB1-specific agonists (see Supplementary Fig. S11) while administering DSS in their drinking water (Fig. 4C). All metrics

of the disease, i.e., weight loss (Fig. 4D), disease activity index (Fig. 4E), histology score (Fig. 4F) and fibrotic shortening of the colon (see *Extended data*; Supplementary Fig. S12A–F) were significantly ameliorated by two PRKAB1-specific agonists, A-769662 (A7) and PF-06409577 (PF), whereas the non-specific AMPK-agonist, Metformin, did not. To obtain proof-of-

**Fig. 4 Validation of PRKAB1-specific agonists in murine models of DSS-induced colitis and human organoid-microbe co-culture studies. A**, **B** Prediction of mouse colitis models relevant to human IBD. Computational workflow (**A**) for the prediction of mouse models of colitis that would be relevant for human IBD using *BoNE*. **B** Table (*left*) and bar plots (*right*) summarizing ROC/AUC analysis of various mouse models of colitis for their ability to model human IBD continuum using C#1-2-3 of the clustered Boolean implication network. Bulk = whole distal colon; epithelium = sorted epithelial cells. **C** Schematic outlining experimental design of DSS colitis model used to test the efficacy of multiple target-specific (PRKAB1-agonists PF-06409577 and A-769662) and non-specific (Metformin) drugs. The pharmacologic properties of each compound is summarized in Supplementary Fig. S11. **D** Line graph showing body weight change monitored daily during the course of acute DSS colitis. **E** Line graph of disease activity index (DAI) using stool consistency (0–4), rectal bleeding (0–4), and weight loss (0–4) as scoring criteria. **F** Scatter plot of histomorphological evaluation of inflammation by H&E stained colon tissues using inflammatory cell infiltrate (1–3), and epithelial architecture (1–3) as scoring criteria. For **D**–**F** $n = 3$–7 mice per group (DSS- $n = 3$, Vehicle $n = 7$, Met $n = 5$, A7 $n = 5$, PF $n = 5$). Extended histologic and immunohistologic analyses of colon tissue is presented in Supplementary Fig. S12. **G** Schematic of the stem cell-based organoid model and the generation of differentiated enteroid-derived monolayers (EDMs) for co-culture studies with microbes. Murine organoid studies are presented in Supplementary Fig. S13. **H** Light micrographs of human 3D enteroids (top row) and immunofluorescent micrographs of the TJ marker ZO-1 on human 2D enteroids (bottom row) isolated from healthy human colons or from colons of actively inflamed UC and CD patients. White arrowheads identify disrupted TJs. Scale bars are 100 μm and 10 μm in light micrographs and fluorescent micrographs, respectively. **I** Bar graph showing quantification of disrupted tri-cellular tight-junctions (TTJs) of healthy ($n = 3$), UC ($n = 3$), and CD ($n = 3$) patient-derived EDMs. **J** Bar graph of maximal absolute TEER values of UC and CD EDMs at ~48 h after EDM preparation when monolayers reach steady-state, as determined by two consecutive TEER reads that are similar ($n = 3$–5 reads/patient-derived organoid lines in 4I). **K–R** Immunofluorescent micrographs (**K**, UC; **L**, CD), a bar graph of quantification of TTJ disruption (**M**, UC; **N**, CD), and a bar graph showing the change in TEER (**O**, UC; **P**, CD) in patient-derived EDMs treated or not, with PF-06409577 (PF) for 16 h. All scale bars are 10 μm. Bar graph showing the change in TEER in UC (**Q**) and CD (**R**) patient EDMs treated, or not, with PF and exposed to LF-82 for 8 h. The frequency of disrupted TJs was calculated from 3 randomly chosen fields. See also Supplementary Fig. S14 for corresponding data in healthy patient-derived organoids. All TEER results were from 3 independent experiments. All data shown as mean ± S.E.M. and one-way ANOVA using Tukey's multiple comparisons test and $p \leq 0.05$ cutoff was used to determine significance. Source data are provided as a Source Data file.

mechanism for effective target (PRKAB1) activation and reversal of epithelial leakiness, we analyzed by IHC the colon tissues for activation of the SPS pathway (the proposed mechanism of action of PRKAB1 in the epithelium) and reduction of levels of claudin-2 (CLDN2). Treatment with PRKAB1-specific agonists not only showed the most prominent activation of the SPS-pathway (as determined by anti-pS$^{245}$GIV; Supplementary Fig. S12G) and near complete reversal of claudin-2 (CLDN2) (Supplementary Fig. S12G), but also showed restoration of goblet cells (PAS staining), and ameliorated fibrosis (Trichrome stain) (Supplementary Fig. S12G). These studies in a DSS-induced colitis model validate the use of PRKAB1-agonists as barrier-protective therapy and provide preclinical proof of concept and mechanism, and that precision targeting of this isoform of AMPK outperformed non-specific agonist Metformin.

*PRKAB1-agonists protect the epithelial barrier in network-rationalized organoid models.* To define the epithelium-specific mechanism of action of PRKAB1-agonists, we used an in vitro enteroid-derived monolayer (EDM) culture system[60], in which stem cells isolated from the colonic crypts of mice are grown as 3D organoids and subsequently plated onto trans-well inserts where they were differentiated into mature colonic epithelium (Fig. 4G). These EDMs are known to contain diverse cell types and maintain a polarized architecture like what is seen in vivo[61], and allow for access to the apical and basolateral compartments and measurement of barrier function via trans-epithelial electrical resistance (TEER) and confocal microscopy. First, using organoids derived from colons of AMPKα1/α2-Villin-Cre KO[62] mice, in which both the catalytic subunits of AMPK are depleted (Supplementary Fig. S13A, B), we confirmed that PRKAB1-agonists require the catalytically active kinase to be able to stabilize the epithelial barrier (Supplementary Fig. S13C) and activate the SPS-pathway in polarized EDMs (Supplementary Fig. S13D). Next, we asked if PRKAB1-agonists can also stabilize/protect the epithelial barrier when exposed to live microbes. Once again, we used *BoNE* to confirm that EDMs infected with pathogenic microbes (*E. coli* and *Shigella*) but not probiotics could serve as models that recapitulate the barrier-defect transcript signature in human IBD (Supplementary Fig. S13E). We

pre-treated murine EDMs with PRKAB1-agonists (A7 and PF; Supplementary Fig. S13F) and then challenged them with adherent invasive *E. coli (AIEC)-LF82*; this strain, originally isolated from a chronic ileal lesion from a CD patient[63]. After 8 h of infection, control (untreated) monolayers showed a 60% reduction in TEER, whereas all PRKAB1-agonist treated conditions showed protection (Supplementary Fig. S13F). Similar results were observed using lipopolysaccharide (LPS), a critical outer-membrane component of gram-negative bacteria (Supplementary Fig. S13G). As expected, decreasing TEER after LF-82 infection was associated with junctional collapse, preferentially at tri-cellular TJs, that was prevented by pretreatment with the PRKAB1-agonist PF-06409577 (Supplementary Fig. S13H, I). Staining for pS245-GIV was observed at junctions exclusively after PF treatment, indicating that the stabilization of TJs via activation of the SPS-pathway may serve as the mechanism of action of PRKAB1-agonists. Thus, PRKAB1-agonists activate the SPS-pathway in gut epithelium and prevent disruption of the intestinal barrier when exposed to luminal stressors such as live microbes (pathogens) or microbial products (LPS).

*PRKAB1-agonists restore the leaky barrier in patient-derived organoids.* To translate findings from mice to humans, and most importantly, to assess the impact of PRKAB1-agonists on the gut barrier of IBD-afflicted patients, we recruited a total of 18 patients (4 healthy, 4 UC and 10 CD; see Table 2), successfully generated organoids and EDMs from their colons (Fig. 4G, H, top panel) and subsequently assessed them for barrier integrity. Barrier integrity, as determined by confocal microscopy on EDMs stained for the TJ-marker ZO1 and assessed for the frequency of disrupted ('burst') tri-cellular TJs (TTJ)/high power field, was impaired in both UC and CD, but not in monolayers prepared from healthy controls (Fig. 4H, I). TEER values were consistently lower in UC and CD EDMs compared to healthy controls (Fig. 4J). Because the diseased organoids maintained what appeared to be an intrinsic defect in the epithelial barrier, we used these as models for testing the efficacy of PRKAB1-agonist PF-06409577 as barrier restorative and/or protective therapy. Treatment of both UC and CD-derived EDMs activated the SPS-pathway (pS245GIV signal; Fig. 4K, L), repaired the 'burst' TTJs

**Table 2 Cohort characteristics for human organoid-based studies [related to Figs. 4 and 5A, Main Text].**

| Internal number (UC San Diego HUMANOID) | % TEER increase after PF, normalized to untreated (Average from 3–5 experiments) | Disease location; Disease behavior | Drug history | Disease duration (years) | Disease score[a] SES |
|---|---|---|---|---|---|
| Healthy H4 | 52.1 | n/a | n/a | n/a | n/a |
| Healthy H9 | 17.5 | n/a | n/a | n/a | n/a |
| Healthy H12 | 9 | n/a | n/a | n/a | n/a |
| Healthy HI3 | 6.8 | n/a | n/a | n/a | n/a |
| UC1 | 30 | Pancolitis | Current Adalimumab | 9 | 8 |
| UC2 | 100 | Left sided colitis | Only ASA | 21 | 9 |
| UC3 | 62 | Left sided colitis | Infliximab (just started) | 3 | 6 |
| UC13 | 187.5 | Proctitis | Vedolizumab | 0.28 | 5 |
| CD2 | 37 | Ileocolitis | Current Remicade | 2.8 | Unk |
| CD3 | 18 | Ileitis | Current Humira | 3 | Unk |
| CD10 | 49.45 | Ileitis; Non-stricturing, non-penetrating | Current Adalimumab, Past infliximab | 23 | 8 |
| CD11 | 55.5 | Ileocolitis; Stricturing | Past Vedolizumab, Past Adalimumab | 19 | 6 |
| CD20 | 47.3 | Ileitis; Non-stricturing, non-penetrating | Current Adalimumab | 1 | 0 |
| CD24 | 15.3 | Colitis; Stricturing | Current Vedolizumab | 6 | 0 |
| CD30 | 377.5 | Ileitis; Non-stricturing, non-penetrating | Naive | 18 | 0 |
| CD21 | 211.4 | Colitis; Non-stricturing; non-penetrating | Current Infliximab | 1 | 8 |
| CD32 | 52.9 | Ileocolitis; Penetrating | Current Infliximab | 20 | 3 |
| CD42 | 3.3 | Ileocolitis; Penetrating | Past Infliximab | 12 | 4 |

In 18 human subjects in the cohort, 11 are Male, 7 are Female from the following age group: below 20 years: 3; 21–40 years: 10; above 40 years: 5. Diseased Cohort (N = 14).
Increase in TEER less than ≤25% N = 3 of 14; 21.42% (all CD).
Increase in TEER within >25–75%, N = 7 of 14; 50% (5 CD, 2 UC).
Increase in TEER is above >75%, N = 4 of 14; 28.57% (2 CD, 2 UC).
Unk unknown; n/a not available.
[a]Disease Score: UC: Histo Geboes Score, CD: SES-CD score/Disease score.

(Fig. 4K–N), with just ~25% increase in TEER across monolayers (Fig. 4O, P). PF-06409577 also demonstrated barrier-protective efficacy when monolayers were challenged with *AIEC-LF82* in both healthy (Supplementary Fig. S14A–C) and IBD-derived EDMs (Fig. 4Q, R).

We next assessed the efficacy of PRKAB1-agonist PF-06409577 using ≥25% increase in TEER as a criterion for the response to barrier-restorative treatment. A majority (~80%) of all diseased organoids responded to treatment with a single dose of 1 µM PF-06409577 (Fig. 5A; Table 2). A multivariate analysis suggested that treatment is effective (*p* < 0.001) in IBD-patient-derived EDMs and that the effect of treatment is not confounded by age, gender, race, prior treatment history, and IBD-disease subtypes (Fig. 5A; Supplementary Fig. S15A–D). Healthy organoids did not show significant changes in TEER. Findings are consistent with

UC- and CD-alone networks (Supplementary Figs. S16 and S17; Supplementary Tables 5 and 6), which predicted that PRKAB1 is poised early in the disease continuum in both subtypes of IBD. Furthermore, the combination of PRKAB1-agonists with anti-inflammatory agents is likely to show therapeutic synergy because they seek to upregulate gene clusters on the healthy side of the network, whereas all other FDA-approved agents seek to suppress the expression of pro-inflammatory genes on the diseased side (Supplementary Fig. S18). These results provide proof-of-concept and mechanism in the human gut lining and demonstrate therapeutic response in a human pre-clinical model.

*Boolean Network Explorer predicts successful* versus *abandoned targets in IBD.* Next we asked if *BoNE* can be exploited to statistically vet the probability of PRKAB1, or any other target, to

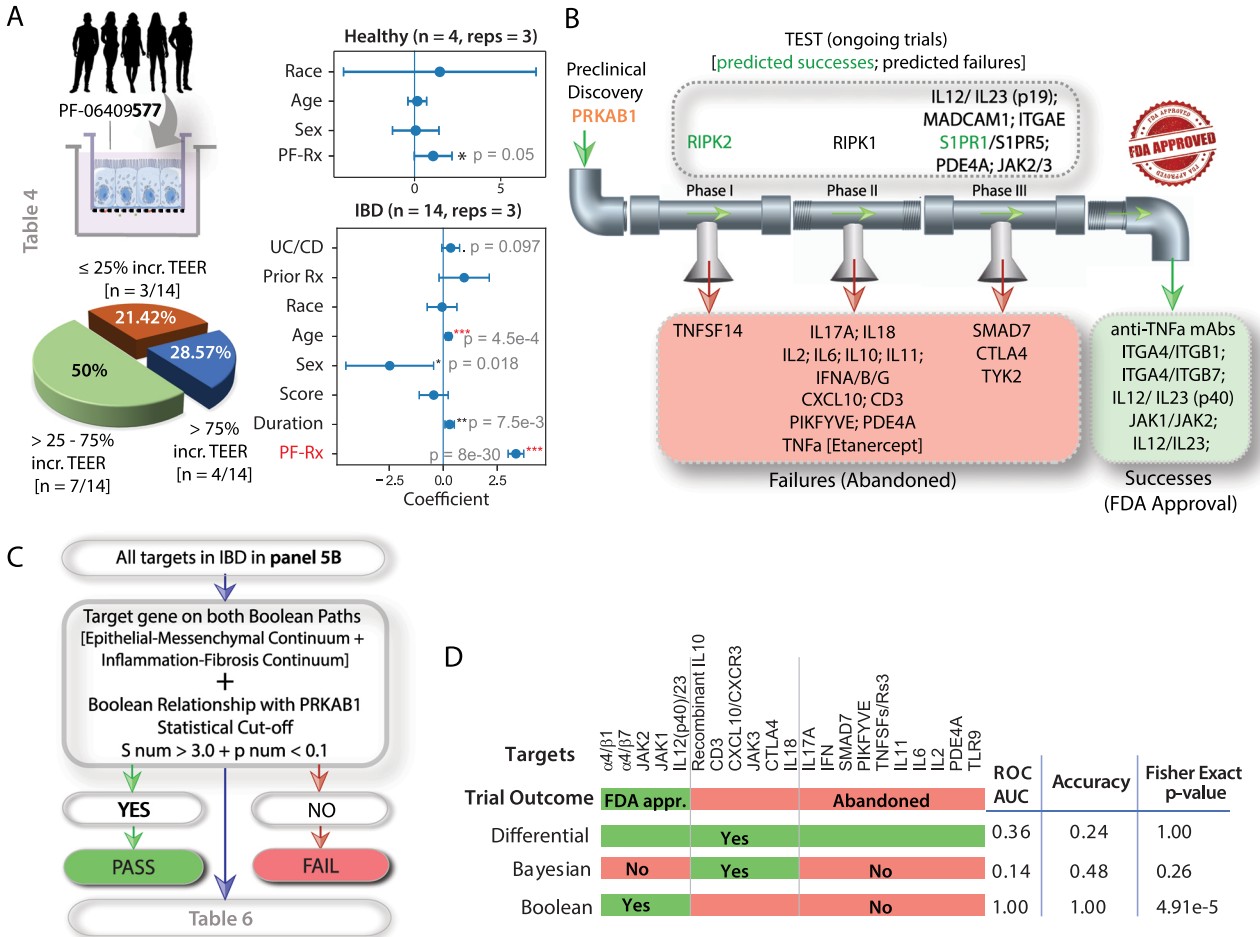

**Fig. 5 Diseased organoids and the Boolean Network Explorer predict clinical therapeutic efficacy in humans. A** *Left top*: Enteroid monolayers from healthy and IBD-afflicted patients were treated with PRKAB1-agonist PF-06409577 (PF-Rx) and assessed for therapeutic response, i.e., >25% increase in TEER [see Table 2 for metadata linked to these patient-derived organoids]. *Left bottom*: Pie chart showing the fraction of IBD-derived EDMs in each category of response. *Right*. Multivariate analysis models the TEER measurement before the treatment as a linear combination of TEER measurement after treatment, age, gender (Female:0, Male:1), race (African American:0, Asian:1, Caucasian:2, Hispanic:3, Middle Eastern:4), treatment history (Naive:0, Infliximab:1, Adalimumab:2, Adalimumab + Infliximab:4, Vedolizumab:3, Vedolizumab + Adalimumab:5). Coefficient of each variables (at the center) with 95% confidence intervals (as error bars) and the *p* values were illustrated in the bar plot. The *p* value for each term tests the null hypothesis that the coefficient is equal to zero (no effect). n number of patients, reps average number of repeats TEER measurements/patient; *\**p* ≤ 0.05; \*\**p* ≤ 0.01; \*\*\**p* ≤ 0.001; \*\*\*\**p* ≤ 0.0001 [see also Supplementary Fig. S15A–D for extended analyses on UC and CD-derived organoids separately, and Supplementary Figs. S16 and 17 for the position of the target PRKAB1 within the Boolean implication networks of UC and CD, respectively]. **B** Drug targets are arranged by their status in clinical trials. Ongoing trials (top); Abandoned trials (red box); FDA-approved drugs that are considered a success (green box). See also Table 3 for the target gene(s) and the outcome of the clinical trials. **C** Computational steps for prediction of success and failure. The targets are checked for consistency in both Epithelial-Mesenchymal and inflammation-fibrosis Boolean paths and a strong PRKAB1 high ≥ X low Boolean implication relationship (*S* > 3, *p* < 0.1). **D** Prediction of outcomes in clinical trials by Boolean analysis compared to Differential and Bayesian analyses [see also Supplementary Fig. S19]. Fisher exact test (two-sided) is performed on a 2 × 2 contingency table based on the prediction. Source data are provided as a Source Data file.

**Table 3 Statistical analysis of the likelihood of FDA approval of targets in IBD [related to Fig. 5B–D].**

| Drug Class | Drug Target | Target Gene | BooleanNet Statistics | | Boolean Paths | Prediction | FDA approval |
|---|---|---|---|---|---|---|---|
| | | | S (>3) | P (<0.1) | | | |
| Anti-traffic | α4/β7 | ITGA4 | 3.535 | 0.084 | Both | PASS | YES |
| | | ITGB7 | 1.664 | 0.296 | Both | | |
| | α4/β1 | ITGB1 | 3.175 | 0.044 | Both | PASS | YES* |
| | Madcam1 | MADCAM1 | 1.336 | 0.310 | None | FAIL | TEST** |
| Checkpoint Inhibitors | PDL1 | CD274 | 4 | 0 | Both | PASS | TEST** |
| | PD1 | PDCD1 | 2.294 | 0.273 | None | FAIL | TEST** |
| | CTLA4 | CTLA4 | 2.683 | 0.225 | Both | FAIL | NO |
| Cytokines | IL2 | IL2 | 2.020 | 0.179 | None | FAIL | NO |
| | | IL2R | 0 | 0 | None | | |
| | IL11 | IL11 | 2.713 | 0.075 | Both | FAIL | NO |
| | | IL11RA | -1.032 | 0.769 | None | | |
| | Recombinant IL10 | IL10 | 3.152 | 0.120 | Both | FAIL (ligand) PASS (IL10RA antagonists) | NO |
| | | IL10RA | 3.175 | 0.038 | Both | | |
| | IL6 | IL6 | 2.910 | 0.151 | None | FAIL | NO |
| | | IL6R | 0.301 | 0.539 | None | | |
| | IL18 | IL18 | **-2.828** | **0.925** | Both | FAIL; predicted exacerbation | NO |
| | | IL18R1 | 3.872 | 0 | Both | | |
| | IL17A | IL17A | 2.523 | 0.225 | None | FAIL; predicted exacerbation | NO |
| | | IL17RA | 1.732 | 0 | None | | |
| | | IL17RC | **-2.474** | **0.909** | Both | | |
| | IL12(p40)/23 | IL12A (p35) | 2.110 | 0.195 | None | PASS © | YES# |
| | | IL12B (p40) | -0.301 | 0.462 | None | | |
| | | IL12RB1 | 3.299 | 0.117 | Both | | |
| | | IL12RB2 | 3.605 | 0 | Both | | |
| | IL12/23 (p19) | IL23A (p19) | 2.218 | 0.185 | None | FAIL ¢ | TEST** |
| | | IL23R | 0 | 0.618 | None | | |
| | TNFSFs/Rs | TNFα (a.k.a., SF1B, SF2) | 1.147 | 0.416 | None | FAIL | NO^c, f |
| | | LTα (a.k.a., SF1A; LTA) | 1.89 | 0.31 | None | | |
| | | TNFRSF1A (TNFR1) | -0.82 | 0.71 | None | BORDERLINE PASS^δ | YES^δ, f |
| | | TNFRSF1B (TNFR2) | 3.1 | 0.11 | Both | | |
| | | TNFRSF14 | **-2.333** | **0.891** | None | FAIL | NO |
| | IFN | IFNG | 2.182 | 0.242 | Both | FAIL | NO |
| | | IFNA1 | 1 | 0 | None | | |
| | | IFNB1 | 1.666 | 0.173 | None | | |
| | | IFNA2 | 1 | 0 | None | | |
| Anti-T cell | CD3 | CD3D | 2.182 | 0.264 | Both | FAIL | NO |
| | | CD3E | 2.672 | 0.155 | Both | | |
| | | CD3G | 3 | 0.127 | Both | | |
| Cell signaling | Kinases | JAK1 | 3.464 | 0 | Both | PASS | YES¶ |
| | | JAK2 | 3.605 | 0 | Both | PASS§ | YES¶ |
| | | JAK3 | 2.752 | 0.200 | Both | FAIL | NO |
| | | RIPK1 | 0.707 | 0.196 | None | FAIL | TEST** |
| | | RIPK2 | 3.162 | 0 | Both | PASS | TEST** |
| | | TYK2 | 1.154 | 0.166 | None | FAIL | TEST** |
| | | PIKFYVE | 2.592 | 0.2125 | Both | FAIL | NO^£ |
| | G protein coupled receptors (GPCRs); cAMP | CXCL10 | 3.605 | 0 | EMT | FAIL | NO |
| | | CXCR3 | 1.889 | 0.114 | None | | |
| | | PDE4A | 2.041 | 0.076 | None | FAIL | NO |
| | | S1PR1 | 4.123 | 0 | Both | PASS | TEST** |
| | | S1PR5 | 2.773 | 0.107 | None | FAIL | TEST** |
| | TLRs | TLR9 | 2.405 | 0.185 | None | FAIL | NO |
| | TGFβ/SMAD pathway | SMAD7 | 0 | 0.518 | None | FAIL | NO |
| | | TGFB1 | 3.098 | 0.105 | Both | FAIL | TEST** |
| | | TGFBR1 | 3.605 | 0 | Both | PASS | TEST** |

Red denotes, S > 3.0 or P < 0.1 or presence on both continuum paths.
Blue denotes strong Boolean relationship (PRKAB1 high = > X high) of genes (or targets) that should not be targeted with antagonists.
[a]Approved but withdrawn later due to side effects.
[b](TEST) Indicates either ongoing Phase III, pending FDA status, or untested targets.
[c]Ustekinumab.
[d]Risankizumab.
[e]Approved for CD, pending approval for UC.
[f]Etanercept (soluble TNFR2 ectodomain), binds both TNFα and LTα.
[g]Indeterminate mechanism of action [direct cytotoxic effect on mAb-coated target cell has been proposed].
[h]Multiple anti-TNFα-specific mAbs that antagonize TNFR1/2; signaling.
[i]More effective in UC than CD.
[j]Tofacitinib (JAK1-3/TYK2 inhibitor) approved for UC, failed in CD; multiple other JAK1/3 orJAK1 inhibitors are in Phase III.
[k]Apilimod mesylate [STA5326]

succeed in clinical trials. The primary source of trial failure has been and remains an inability to demonstrate efficacy[64]; many drugs that were effective in inbred mice lacked efficacy in heterogeneous cohorts of patients. A comprehensive review of the literature identified five FDA-approved drugs, sixteen drug targets that were abandoned at different phases (I, II or III) in clinical trials, and seven currently ongoing trials (Fig. 5B). We set a criterion that effective targets must appear on both Boolean paths (EMT and inflammation/fibrosis; Fig. 2C, D). To make this process stringent, an additional criterion was included, i.e., it must have a strong relationship with the target (PRKAB1), meeting/exceeding the *BooleanNet* statistical threshold $SThr > 3$ and $pThr < 0.1$[65]; (Fig. 5C; Table 3). *BoNE* successfully distinguished the FDA-approved *vs.* the abandoned targets (ROC AUC 1.00; Accuracy 1.00; Fig. 5D; Supplementary Fig. S19). By contrast, all targets were significant by differential analysis (high false-positive rate; Fig. 5D; Supplementary Fig. S19) and almost all the 'successes' were missed by Bayesian analysis (high false-negative rate; Fig. 5D; Supplementary Fig. S19A, B). Furthermore, a Boolean association analysis among targets (FDA-approved and failed) was carried out to see if targets tend to implicate each other. We used equivalent instead of high/low or opposite because these targets are all anti-inflammatory so they should be positively associated with each other. FDA-approved targets were found to overwhelmingly implicate each other (Supplementary Fig. S19C), whereas most of the abandoned targets do not implicate any of the approved targets. Findings indicate that *BoNE* can accurately assess the probability of a target passing an efficacy test in Phase III clinical trials. Given the retrospective nature of this analysis, these findings need to be confirmed within the framework of other randomized clinical trials, in conjunction with large-scale transcriptomic studies before *BoNE* can be used to pick targets in IBD therapeutics.

In conclusion, despite being at the forefront of biomedical research, therapies that can restore and/or protect the integrity of the gut barrier in IBD had not emerged. We have addressed this unmet need using an AI-guided drug discovery approach that differs from the current practice in three fundamental ways: 1) Target identification and prediction modeling that is guided by a Boolean implication network; 2) Target validation in network-rationalized animal models that most accurately recapitulate the human disease; 3) Target validation in human preclinical organoid co-culture models, inspiring the concept of Phase '0' trials that have the potential to personalize the choice of therapies. The combined synergy of these approaches validates a first-in-class agent in addressing the broken gut barrier in IBD.

## Methods

Detailed methods for computational modeling, AI-guided target identification and target validation in murine models and patient-derived organoid co-cultures is presented in Supplementary Online Materials, and mentioned in brief here.

**Computational approach**. An overview of the key approach is shown in Fig. 1. Modeling continuum states in IBD was performed using Boolean Network Explorer (*BoNE*)[66]. We created an asymmetric gene expression network of IBD using a computational method based on Boolean logic[30,65,67]. To build the network, we analyzed two publicly available colon-derived transcriptomic datasets from IBD patients, GSE83687[17] and GSE7366135 (see Supplementary Data 1). These two datasets ('test cohorts') were independently analyzed and kept separate from each other at all times. A Boolean Network Explorer (*BoNE*; see Supplementary Methods) computational tool was introduced, which uses asymmetric properties of Boolean implication relationships (BIRs)[65]; to model natural progressive time-series changes in major cellular compartments that initiate, propagate and perpetuate inflammation in IBD and are likely to be important for disease progression. *BoNE* provides an integrated platform for the construction, visualization and querying of a network of progressive changes much like a disease map (in this case, IBD-map) in three steps: First, the expression levels of all genes in these datasets were converted to binary values (high or low) using the *StepMiner* algorithm[67]. Second, gene expression relationships between pairs of genes were classified into one-of-six possible BIRs, two symmetric and four asymmetric, and

expressed as Boolean implication statements (Fig. 2A). This offers a distinct advantage from currently used conventional computational methods that rely exclusively on symmetric linear relationships from gene expression data, e.g, Differential, correlation-network, coexpression-network, mutual information-network, and the Bayesian signature that was originally identified using one of the test cohorts, GSE83687[17]. The other advantage of using BIRs is that they are robust to the noise of sample heterogeneity (i.e., healthy, diseased, genotypic, phenotypic, ethnic, interventions, disease severity) and every sample follows the same mathematical equation, and hence is likely to be reproducible in independent validation datasets. The heterogeneity of samples in each of the datasets used in this study is highlighted in Supplementary Data 1. Third, genes with similar expression architectures, determined by sharing at least half of the equivalences among gene pairs, were grouped into clusters and organized into a network by determining the overwhelming Boolean relationships observed between any two clusters[30,65] (Fig. 2A). In the resultant Boolean implication network, clusters of genes are the nodes, and the BIR between the clusters are the directed edges; *BoNE* enables their discovery in an unsupervised way while remaining agnostic to the sample type. All gene expression datasets were visualized using Hierarchical Exploration of Gene Expression Microarrys Online (HEGEMON) framework[68].

**Animal studies**. For in vivo animal experiments, the experiments (including animal breeding, housing, DSS treatment and euthanize) were performed according to the University of California San Diego Institutional Animal Care and Use Committee (IACUC) policies under the animal protocol numbers S18086 and S17223. All methods were carried out in accordance with relevant guidelines and regulations and the experimental protocols were approved by institutional policies and reviewed by the licensing committee. Intestinal crypts were isolated either from the proximal and the mid-colon of WT C57BL/6 or AMPK KO mice; generated from gender- and age-matched littermates of age 5–7 weeks. For DSS-colitis experiments, 7–8-wk old C57Bl/6 mice were obtained from Jackson Laboratories (Bay Harbor, ME). Animals were bred, housed (light and dark cycle of 12 h each, humidity 30–70% and room temperature controlled between 68–75 °F). All animals were assessed routinely for signs of pain, suffering and distress associated with procedures. Euthanasia is performed by placing the animals in an equipment where the air was displaced with $CO_2$ at a rate ~50% of the chamber volume/min. The flow was maintained for at least 1 min after respiratory arrest and then the cervical dislocation was performed.

**Human subjects**. For generating healthy and IBD patient-derived organoids, patients were enrolled for colonoscopy as part of routine care for the management of IBD from the University of California, San Diego IBD-Center, following a research protocol compliant with the Human Research Protection Program (HRPP) and approved by the Institutional Review Board (Project ID# 1132632). Healthy colon samples were collected from patients presenting for screening colonoscopy or undergoing the procedure for making the diagnosis of irritable bowel syndrome. Each participant provided a signed informed consent to allow for the collection of colonic tissue biopsies for research purposes to generate 3D organoids. Isolation and biobanking of organoids from these colonic biopsies were carried out using an approved IRB (Project ID # 190105: PI Ghosh and Das) that covers human subject research at the UC San Diego HUMANOID Center of Research Excellence (CoRE). For all the deidentified human subjects, information including age, gender, and previous history of the disease, was collected from the chart following the rules of HIPAA. The study design and the use of human study participants was conducted in accordance to the criteria set by the Declaration of Helsinki.

**Statistics and reproducibility**. Each staining experiments and the claims representing the findings were reproduced in at least 3–5 independent repeats. Gene signature was validated in 15 independent publicly available datasets.

**Reporting summary**. Further information on research design is available in the Nature Research Reporting Summary linked to this article.

## Data availability

Source data are provided with this paper. Publicly available datasets used: GSE83687, GSE73661, GSE6731, E-MTAB–7604, GSE75214, GSE59071, GSE48958, GSE16879, GSE37283, GSE109142, GSE97012, GSE95437, GSE95095, GSE100833, GSE83550, All data is available in the main text or the supplementary materials. Source data are provided with this paper.

## Code availability

The software codes are publicly available at the following links: https://github.com/sahoo00/BoNE[66] https://github.com/sahoo00/Hegemon[68]

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

## Acknowledgements

This work was supported by National Institutes for Health (NIH) grants AI141630 (to P. G.), DK107585, R56 AG069689 and DiaComp Pilot and Feasibility award (to S.D.), R00-CA151673, R01-GM138385, Padres Pedal the Cause/C3 Collaborative Translational Cancer Research Award (San Diego NCI Cancer Centers Council [C3] #PTC2017) (to D. S.). P.G., S.D., and D.S. were also supported by the Leona M. and Harry B. Helmsley Charitable Trust and the NIH (UG3TR003355, UG3TR002968 and R01-AI55696). G.D. K. was supported through The American Association of Immunologists Intersect Fellowship Program for Computational Scientists and Immunologists. Y.M. and L.S. were supported by National Institutes for Health (NIH) training grant (T32 DK 007202). Y.M. was also supported by an NIH CTSA-funded career-development award (1TL1TR001443). S.R.I. was supported by the postdoctoral fellowship grant from NIH (3R01DK107585–02S1).

## Author contributions

D.S., S.D. and P.G. conceptualized, supervised, administered the project and acquired funding to support it. D.S., L.S., I.I.S., G.K., B.L., Y.M., S.R.C., R.P., C.T., M.F., D.L.S., W.J.S, S.D., P.G. were involved in data curation and formal analysis. Computational modeling were carried out by D.S. Animal models of colitis were conducted by L.S. and G.D.K. Immunohistochemical and qPCR studies on patient tissues were carried out by I.M.S., S.R.C., and Y.M. Organoid isolation, culture and their use in experiments were carried out by I.M.S., L.S., R.P., C.T., M.F. and D.L.S., all supervised by S.D. W.J.S. and J.T.C. provided key resources for human subjects and animal studies, respectively. D.S. provided software. D.S., L.S., I.M.S., S.D., and P.G. prepared figures for data visualization, wrote original draft. D.S., S.D., P.G., L.S., I.M.S., and G.D.K. reviewed and edited the draft. All co-authors approved the final version of the manuscript.

## Competing interests

S.D., D.S. and P.G. have a patent on the methodology. Barring this, all authors declare no competing interests.
