## [Peer Review File · Nature Communications]

Reviewers' Comments:

Reviewer #1:

Remarks to the Author:

In their manuscript, "AI-guided discovery of a barrier-protective therapy in inflammatory bowel disease", Sahoo et al. develop multi-step process to identify and prioritize targets that are predicted to shift molecular states from disease causes to normal: 1) identify targets using a Boolean implication network algorithm with some bioinformatic processing steps; and 2) validate identified targets using experimental models that are informed by the constructed networks. Their approach was applied to IBD where they identified targets and identified one to push forward into experimental validation. They also demonstrate the utility of their model in predicting whether a target is likely to make it through phase III trials.

Certainly I am in agreement with this paper on the need to use network-driven approaches to better organize large-scale molecular data into network models that can help order and predict the most appropriate points for therapeutic intervention so that more informed, data-driven hypotheses are generated based on a far more comprehensive consideration of the data. I think the kind of approach the authors describe is along the exact right lines, and some of the results well reflect this, given the ability to discriminate among meaningful healthy, disease and drug response states. However, I have a number of concerns that can be broken into two types: 1) the material presented not being presented in comprehensive enough or clear enough form to fully understand what was done and how the results were obtained (to the level of understanding how to replicate them); and 2) what the network model actually reflects and whether the insights derived from the model were novel enough to highlight the utility of the approach with respect to a completely data-driven drug discovery process. Comments relating to these concerns are spelled out below in the specific comments.

Specific Comments:

1. Authors reference different "network" approaches developed that have ushered in network medicine, references regression-based approaches and coexpression networks, but then indicate that nothing in drug discovery has been done with networks to indicate drugs that would reset disease network. I don't believe this is exactly right as probabilistic causal network models have definitely been developed to identify and prioritize target and repurpose drugs that are predicted by the network to reset disease. There are many examples I believe, but to get the authors exploring, here are a few examples, one of which the authors cited: Cell 153, 707-720; Nat Commun 7, 12092; Nat Genet 49, 1437-1449; Cell Rep 9, 1417-1429. The authors should better position their work in this context. I do think there are some advances here on top of what others have done in this arena, but it should be more clearly spelled out.

2. Authors claim in last paragraph on page 3 that current IBD treatments realize only 30-40% response rate. They cite a paper that is 8 years old (ref 16). There are newer drugs since this time that have had significant benefit to patients. Drugs such as Stelara for example. Current estimates are response rates closer to 60-80%. See for example: Ther Adv Gastroenterol 2019, Vol. 12: 1-14. As above, authors should take the time to characterize their work in a more current context.

3. In the BoNE section of the Methods section the authors claim that their approach discretizing variables (gene expression traits) into two states (high or low) and then classifying the relationships between pairs of variables after application of their Boolean algorithm into one of six types. But nothing is said about why their approach has any advantage over other Bayesian network approaches that also discretize input data into any number of states (for example, in one of the papers cited by the authors, ref 27, the network approached developed by Zhu et al. discretizes into 3 states and similar types of inferences made here can be drawn from the edges in such networks as well). A familiar theme at this point, where the authors just haven't done enough to place their approach into the context of what others have done, and then clearly articulate why what they have done offers an advantage. Usually with such methodologic claims comes direct comparisons to alternative approaches already in use to establish the advantage. If the authors want to make the claim, they should show it.

4. The description of the network construction and identification of BIRs is confusing. Figure 2A makes it appear that the expression are input into a network construction process that delivers Boolean networks and then BIRs are identified from the networks. But the description on page 4 in the methods indicates first step is discretization, second step is classifying relationships between gene pairs, and third building a network by clustering gene pairs. The supp material then tries to clarify that it is the discretization process that delivers the BIRs, which makes Figure 2A appear misleading. The authors need to describe this with enough clarity to enable the results to be replicated.

5. It would seem critical to fully describe the datasets the authors are using to build their networks since it forms the basis of everything they infer. But the authors only point to references, and it appears they point to incorrect references, so again confusing what was done, and then the data being used, how comparable the data are, what states the samples from which the data were derived were in, and so on, not at all described, so really impossible to assess what they have coming out of those sets.

As an example on the confusion, the authors point to references 26 and 27 as the source of the datasets. Reference 26 is to a paper by Arijs et al. in Plos One from 2009. Then in the supp the authors point to GSE73661, which is from a 2018 paper by Arijs et al. (ref 32). Does GSE73661 include data from both papers? So they were both used?

Regarding comparability of the two datasets used to build the networks, for the Peters 2017 set, the authors point to GSE83687 and indicate it was comprised of 60 controls, 32 UC, and 42 Crohn's disease samples? They indicate an $n = 143$, but it looks like 134 samples? Where does the 143 come from? And in this dataset they are comprised of UC and Crohn's patients; are the authors then just looking at networks that reflect common parts of those diseases? Are these distinct diseases or the same? Do they occur in the same part of the intestines? If they are in different parts is the assumption that the molecular mechanisms are the same regardless of location? And then the controls were actually not health but rather were colon cancer patients? And given the IBD patients in this cohort were undergoing surgical resections, they seem to represent the most severe forms of IBD not responding to any treatments. This is in contrast to the Arijs 2018 dataset, which seems to have focused only on UC patients and colon biopsies, with perhaps a more appropriate control group (those undergoing colonoscopy, so more healthy, although perhaps older?) The description of GSE73661 is 120 samples from 44 UC patients and then 12 non-IBD samples and 23 more biopsies from UC patients before IFX treatment? And the other 120 were on a different treatment protocol over time? The authors indicate 178 samples, but again not clear what these all are and without spending a day digging into GSE73661 and GSE83687 datasets (I can't burn days as a reviewer trying to figure out what was done), it is simply not easily possible to figure out what exactly was used here.

There is zero description or discussion from the authors on these different samples and why they could all be combined together, what impact age differences played with controls, what role drug treatments played since Arijs samples were involved in clinical trials, what role severity played given the severity of the Peterson 2017 samples, what role the different intestinal regions played given there was a mix, what role biopsies from inflamed vs. non-inflamed played in the signal, and so on and so forth. In the end, it is difficult to understand what is actually being discriminated here.

6. For identification of BIRs, the authors indicate they used BooleanNet statistics and then indicate $S > 2.5$ and $p < 0.1$ as the thresholds, without any description on what these statistics are, how the FDR was computed and so on.

7. Regarding the construction of the Boolean implication network, the authors claim on page 3 of supp that "BIR is strong and robust when the sample sizes are usually more than 200." No reference or support of this claim is given, because as indicated above, the samples being used are so varied, across so many conditions (age, severity, tissue type, disease type, etc.) that it simply can't be true that some general simulation-based result that indicates BIR are strong with $N > 200$ carries over into this setting. Despite this more varied setting and that the number of samples is < 200 , there is nothing shown that would indicate the results as being "strong and robust".

8. For the clustered BINs, it is not clear if this is a new procedure or if it has been better vetted in other publications. This matters because the authors describe a procedure that conceptually makes sense, but where the selection of thresholds used to actually carry out the clustering seem very arbitrary, with no sensitivity analyses provided that would indicate how the results may vary if different thresholds were used. For example, on page 4 of the supp, the authors describe an approach to ensure internal consistency in the resulting clustered BIN (e.g., avoiding cases in which two genes that are opposite of each other become part of the same connected component) that involves computing the minimal spanning tree for the CBIN, and then computing Jaccard similarity coefficients for all edges, and then rationalizing that if two members share less than half their total individual connections (Jaccard similarity coeff < 0.5), they should be dropped as they are potentially internally inconsistent. But why 0.5, why not 0.7, or 0.8, or 0.9, or 0.4, etc.?

9. In trying to understand the network results described in the first section of the results, the referenced figures 2 and S1 did not seem to depict a network, but rather provide some schematics (as in Figure 2B) and then some clustering results and then performance metrics, and I could not find Table 1 (there was a Table 1 legend in the supp, but I could not find the table anywhere in the submission – perhaps I missed it, but certainly it was not obviously placed in the main text, the supp, or in the zip file I downloaded from the NCOMM site). So basically I have no way of actually understanding the result.

10. In first paragraph of the results the authors claim that using reactome pathway analysis on clusters that emerge from the BIN, reveal the biological processes that the clusters *control*. How can one infer from the clusters that they are actually controlling the biological processes? What is actually meant by that?

11. What does this network actually reflect? Is it some average over all of the samples that were used to construct the network? The authors indicate in the first paragraph of the results and the supp on “discovery of paths...” that they get a time series of the biological processes starting with healthy through to severe disease. I do not see how this could be possibly by looking at Boolean paths from a network constructed by simultaneous consideration of varied samples. I would agree there may be ways to identify clusters that are reflecting health and disease states given those states are well enough annotated in the samples (you have control and disease individuals), but then for disease progression, you have such varied conditions, different diseases in the case of Peterson dataset (UC and Crohn’s that were characterized using different tissue types), different drug treatments, etc. There is absolutely no way to distinguish between time-based disease progression and differences in states among the different samples used to construct the network. Or if there is, that needs to be very clearly articulated and it simply is not in the supp (Boolean paths are explore which just represent paths through the network and again what exactly the network represents is not really discussed, but certainly it does not naturally represent a time series).

12. In second section of results, page 5, the authors talk about distinguishing between IBD and “healthy” using machine learning applied to the gene clusters. But the “healthy” controls are not necessarily healthy given in the case of Peterson they have colon cancer. Further, there is likely severe confounding of the controls given their non-IBD state is likely highly correlated with age (given in the Arijs 2018 controls are undergoing colonoscopy, which is generally only done when you are older). Thus, how do you know you are not picking up signal that reflects cancer given there are well known “field” effects induced by a tumor in normal tissues? How do you know you are not simply picking up an age signature? There are independent validation datasets that do seem to support the authors having a meaningful classifier, but again nothing is made easy for the reviewers, there is no description of the datasets but rather just GSE identifiers are given. The authors need to make a table that describes each of the datasets used, what comprised these datasets (tissues profiled, technology used to profile, what diseases were represented, ages, treatment conditions, etc.)

The authors also don’t say much of anything about what the actual machine learning algorithms were that were used to build the classifiers, other than brief mentions of Boolean, Bayesian and differential approaches in Figure 2F). There are hundreds, maybe even thousands of machine

learning algorithms, and how these algorithms are trained and then tested matters a great deal in understanding the results. It is impossible as far as I can see in the paper to in any way repeat what the authors did here, since they just mention some methods and give some results of their performance in the figures, without actually describing how the methods were fit, etc. There is a section in the supp that describes "measurement of classification strength or prediction accuracy", but nothing on the different classifiers that were built, apparently there were several to distinguish health from disease, response from non-response, active from inactive disease, etc. Are the 7 validation cohorts used seem to focus just on normal vs. IBD, but then there are claims of predicting these other states as well. It is of note that the precision and F1 scores for the normals are generally not very good, whereas the classification to IBD state seems generally good, although cases where it fails (validation 6), but none of this is discussed.

13. For the identification of targets, other than membership in the clusters, the identification was not really carried out in a very network-directed way as promised in the intro and discussion, but rather the authors looked at membership in the C#1-2-3 clusters and matched genes in those to "druggable classes of receptors, enzymes and signal transducers. They identified 17 targets annotated as "response to stress", and then focused on the kinases and in particular the one kinase that had existing specific and potent agonists available, PRKAB1 (AMPK). There was no prioritization based on what the network would have predicted the impact would be on any of the members of these clusters. Do the authors think they would all have the same impact? Is the claim they are all causal? This is not really described.

14. Regarding the proposed mechanism of PRKAB1 in the gut lining, what the authors proposed on page 7 of the results, appears to already be known? The authors indicate at end of first paragraph on page 8 that their algorithm predicts for the first time that PRKAB1-agonists would "work through upholding epithelial polarity and TJ integrity, which in turn should reduce inflammation." And papers such as *Front Pharmacol.* 2018 Jul 16;9:761 show the same kind of thing, where this paper indicates that AMPK activation maintains the structure of tight junctions in colonic epithelial cells, reducing inflammation, reducing DSS damage, etc. Not to say that the authors model does not support this exactly, but at least how described, that the identification of PRKAB1 was not totally data driven from the model the authors constructed, but rather met some criteria on what might make a good target and then the story around that was more or less fit to what is known.

15. Given the authors focus on a target that is already somewhat well known and in particular with respect to the IBD hypotheses derived by this manuscript, why not apply this approach more objectively to identification of all targets, the hypotheses relating to mechanism that obtain given the models and annotations, and then assess both positive and negative accuracy with respect to appropriately recapitulating what is known about current therapeutics that worked versus those that failed. The authors present something pretty intriguing along these lines as detailed in figure 5, but then this was only done in a "guilt-by-association" way, where if connectivity to ampk was shown, then positive, else negative. I think one of the more interesting aspects of the paper. Question is whether other of the targets for FDA approved drugs would have been similarly uncovered like PRKAB1 and then in turn implicated the others that were FDA approved, whether the biology uncovered was consistent with what was known, etc. I think this would make the result far more powerful.

16. Regarding the network-rationalized murine model selection process, the question would be if you did not use the network model at all, but instead took a look at what is known today on ampk, processes relating to IBD it is involved in, known role agonists play in those processes, would you have come up with the same mouse model? Or would that have led you astray and the network would have corrected the thinking?

Reviewer #2:

Remarks to the Author:

This is a well performed extensive study that uses computational methods to identify new targets to treat IBD. This approach potentially identifies prediction models to be used at early stages in the clinical pipeline. In addition, the methods described are used to identify the best preclinical

models to test the validity of the target.

Overall the study is well designed, and the manuscript well written.

I have a few comments however I would like the authors to address/respond to.

The training and testing cohorts include quite heterogeneous populations. For instance, GSE83687 contains samples from different locations (small bowel including but not exclusively ileum, rectum, sigmoid colon, and right colon). Moreover, both Crohn's disease and ulcerative colitis samples are analyzed together. How did the authors correct for these variables (i.e. disease location, disease type)? Given the marked differences in the transcriptomes of different intestinal locations, transcriptional analysis of these type of samples is normally performed separately. However, in the case of network analysis it is not clear from the manuscript how this was accounted for and how it could have affected the results. Also, while UC and CD share some mechanisms, epithelial dysfunction for instance is one of the main differential pathways between the two diseases. I would predict that had the authors analyzed UC and CD samples separately, the results of their Boolean analysis may have been somehow different. Did the authors perform any analysis on more homogeneous groups of samples (i.e. colonic UC only, colonic CD only or SB CD only).

The findings on the superiority of the Boolean analysis to predict response to different treatments (Fig 2F-G) are interesting. Was the C1-3 signature the only one that consistently segregate active from inactive as well as R to NR to both IFX and vedo? How do the authors explain this consistency? Could this prediction of response be related to the severity of the disease (more severe patients tend to respond worse to treatment)? The fact that response to two drugs that target independent MOA can be predicted by the same signature that is also the same that segregates by disease activity would suggest that. It would be interesting though to hear what the authors think about that.

The authors state in page 8 that "the algorithm is unable to pick the direction, start and end of these events. Our analysis...barrier disruption". Please explain how this is not in contradiction of Figure 2 C.

Could the authors comment on the findings on Suppl. Figure 6 (cohort 2), where PRKAB1 expression was maintained lower than healthy mucosa in remitting patients? This seems to contradict their statement (page 8) that "low levels of PRKAB1 correlates with a higher degree of leakiness of the epithelial barrier...". This correlation is true in active disease, but it appears that inactive disease maintains low levels of PRKAB1 expression in the absence of higher CLDN2 or proinflammatory cytokine expression. Could the authors hypothesize what the implications of this persistent low PRKAB1 expression could be on disease progression. Would the use of specific activators of PRKAB1 could also benefit maintenance of remission?

Please revise the images chosen in Figure S9 for active (mod to severe) UC. Specially the second mod to severe UC sample (CLDN2 staining) shows typical architectural crypt disruption commonly see in involved UC samples, but lacks signs of acute disease (epithelial loss, and lamina propria increased infiltrate).

TEER measurements should be provided at different points over time for each culture, as they are severely influenced by the number of cells plated, their proliferating capacity and the plate confluency. Indeed, the authors mention in the suppl methods that resistance was measured at 1h intervals for 8 h. How was the time point for Figures 4 I chosen? Also, what was the n for those experiments? Please state in the figure legend.

Where were biopsies taken from to generate organoids (active disease? Involved areas?)

In figure 5A the effect of other variables such as endoscopic disease activity, time since diagnosis, treatment at the time of endoscopy or intestinal segment from which biopsies are obtained to generate organoids, needs to be tested.

Please describe how is "aggressiveness of disease" (page 6) defined.

The statement on page 11 "combination of...therapeutic signaling" is not clear what it is based on. To my understanding the authors do not provide any data to back that up. That would definitely be a possibility worth addressing though.

Reviewer #3:

Remarks to the Author:

The authors used an artificial intelligence (AI)-derived process to discover a potential new therapy for inflammatory bowel disease (IBD). The study is large and addresses the biological complexity of IBD using Boolean modeling. However, a key challenge in Boolean modeling is to match the

right model with the right questions in a specific research context. The work is novel and rich in data, but a substantial number of concerns are identified and described below:

1. The authors used and integrated data from two very different and heterogeneous datasets derived from Arijs et al. and Peters et al. published studies. This analysis has the following issues:

a. Arijs et al. RNA dataset is derived from Affymetrix microarray analysis while Peters et al. dataset is derived from RNA-sequencing analysis.

b. Arijs et al. RNA samples were extracted from mucosal biopsies, while Peters et al. were extracted from surgical specimens.

c. Arijs et al. control samples derived from healthy individuals, while Peters et al. Peters et al. derived from colon cancer patients (10cm away from tumor site).

d. Arijs et al. dataset derives from patients who are refractory to corticosteroids and/or immunosuppression, while Peters et al. dataset derives from patients treated with different treatments.

The authors did not evaluate these clinical differences into their datasets and integrated all gene expression data.

2. The authors present in figure 2C, a map of longitudinal gene network analysis from healthy to IBD. This graph shows that tight junction network is the one deregulated early in the IBD development process followed by bioenergetics changes and then immune-related events. This analysis is only appropriate and valid if RNA datasets derived from patient biopsy samples collected in different time points during IBD development, treatment and remission in the same patient samples.

3. In Figure 3A, the authors show that 336 druggable gene targets and 62 genes related to responses to stimuli. A subset (n=17) of these 62 genes are related to the stress response. How the authors decided to focus on these 17 stress-related genes. Protein-protein network analysis identified that this 17-gene signature is a key regulator of all these genes? Also, how the authors focused their interest on PRKAB1 and not in the other 16 gene targets, what is their rationale?

4. The authors show that PRKAB1 is a novel target and its mRNA levels in colonic biopsies are decreased relative to controls. First, the number of samples (n=9) is very low to justify the identification of a new clinically-relevant drug target. Second, the authors do not show PRKAB1 mRNA levels in UC relative to CD colonic biopsies and how its expression correlates with different disease location (ileum vs colon) and drug treatment. There is extensive literature showing that IBD disease location is a key factor affecting gene expression patterns.

5. The identification of the PRKAB1/AMPK pathway in IBD is not novel; there is previous literature related to the role of this pathway in IBD development and intestinal epithelial permeability. In 2010, Bai et al. showed that activation of this pathway by using an AMPK agonist suppresses the inflammatory response in a TNBS-induced colitis mouse model (Bai et al. *Biochem Pharmacol.* 2010 Dec 1;80(11):1708-17). Sun X et al. showed that AMPK activation enhances intestinal epithelial barrier function (Sun X et al. *Cell Death Differ.* 2017 May;24(5):819-831). Furthermore, Chen L. et al. shows that AMPK activation promotes the expression and assembly of tight junctions (Chen L et al. *Front Pharmacol.* 2018 Jul 16;9:761).

REVIEWER #1:

General Comments: *In their manuscript, “AI-guided discovery of a barrier-protective therapy in inflammatory bowel disease”, Sahoo et al. develop multi-step process to identify and prioritize targets that are predicted to shift molecular states from disease causes to normal: 1) identify targets using a Boolean implication network algorithm with some bioinformatic processing steps; and 2) validate identified targets using experimental models that are informed by the constructed networks. Their approach was applied to IBD where they identified targets and identified one to push forward into experimental validation. They also demonstrate the utility of their model in predicting whether a target is likely to make it through phase III trials.*

Certainly ***I am in agreement with this paper on the need to use network-driven approaches*** to better organize large-scale molecular data into network models that can ***help order and predict*** the most appropriate points for therapeutic intervention so that more informed, ***data-driven hypotheses*** are generated based on a far more comprehensive consideration of the data. ***I think the kind of approach the authors describe is along the exact right lines, and some of the results well reflect this***, given the ability to discriminate among meaningful healthy, disease and drug response states.

However, I have a number of concerns that can be broken into two types: 1) the material presented not being presented in comprehensive enough or clear enough form to fully understand what was done and how the results were obtained (to the level of understanding how to replicate them); and 2) what the network model actually reflects and whether the insights derived from the model were novel enough to highlight the utility of the approach with respect to a completely data-driven drug discovery process. Comments relating to these concerns are spelled out below in the specific comments.

Response to general comments: *We appreciate the generally positive comments regarding the impact of our work (network-based approaches that help order and predict...[generate] data driven hypotheses....based on far more comprehensive consideration of data). We were pleased to see that reviewer #1 found “some of our results reflect...” such approaches. However, he/she also points to two main concerns: 1) materials presented lack comprehensiveness at a level that allows reproducibility; 2) our failure to explain what the model actually reflects so that reader can assess how data driven discovery was actually made and the novelty of such discovery.*

We agree. Brevity in writing and condensed presentation of a large amount of work/data (all compressed in the main text and 100+ pages of Supplementary Materials) likely compromised clarity and comprehensive explanation within the main text.

We appreciate the reviewer taking the time/effort to make comments throughout, pointing out through helpful suggestions how to improve the manuscript. Outlined below is how we have tried to address each of those comments and suggestions.

Specific Comments:

Comment #1. Authors reference different “network” approaches developed that have ushered in network medicine, references regression-based approaches and coexpression networks, but then indicate that nothing in drug discovery has been done with networks to indicate drugs that would reset disease network. I don't believe this is exactly right as probabilistic causal network models have definitely been developed to identify and prioritize target and repurpose drugs that are predicted by the network to reset disease. There are many examples I believe, but to get the authors exploring, here are a few examples, one of which the authors cited: Cell 153, 707-720; Nat Commun 7, 12092; Nat Genet 49, 1437-1449; Cell Rep 9, 1417-1429. The authors

should better position their work in this context. I do think there are some advances here on top of what others have done in this arena, but it should be more clearly spelled out.

Response #1: *We agree. In the original submission, we ensured that we correctly framed the existing art of network-based drug repositioning, prediction of side effects, and target discovery; but we had missed some others. These omissions were not intended to belittle the contributions of, or offend, the pioneers who have come before us. We also want to thank the reviewer for recognizing the novelty of our work, which “presents more advances on top of what others have done in this area”.*

Specific actions taken: *In this revised version, we have added the citations suggested, as well as other new citations [highlighted lines #96-99 on Page #3].*

Comment #2. Authors claim in last paragraph on page 3 that current IBD treatments realize only 30-40% response rate. They cite a paper that is 8 years old (ref 16). There are newer drugs since this time that have had significant benefit to patients. Drugs such as Stelara for example. Current estimates are response rates closer to 60-80%. See for example: *Ther Adv Gastroenterol* 2019, Vol. 12: 1–14. As above, authors should take the time to characterize their work in a more current context.

Response #2: *We have updated our references to the current context. To do so, we used a 2020 comprehensive metanalysis(1). We wanted to be careful (avoid citing one-off papers) because a blanket statement with a % range of efficacy without clarifying whether it was clinical vs endoscopic remission, or what was placebo effect, may be too sweeping to be meaningful. This article has carefully annotated these aspects as below--*

As First line Rx: Clinical Remission: With an estimated placebo rate of achieving remission of 10% in included trials, we anticipate that 31.1%, 17.7%, 23.7%, 22.0%, 19.1%, and 18.5% of infliximab-, adalimumab-, golimumab-, vedolizumab-, tofacitinib-, and ustekinumab treated patients, respectively, would achieve induction of remission.

Endoscopic Remission: With an estimated placebo rate of achieving endoscopic improvement of 30% in induction trials, we estimated that 58.7%, 40.4%, 42.7%, 51.9%, 46.5%, and 44.4% of infliximab-, adalimumab, golimumab-, vedolizumab-, tofacitinib-, and ustekinumab-treated patients, respectively.

As Second line Rx:

Clinical Remission: With an estimated placebo rate of achieving clinical remission of 3% in included trials, we estimated that 3.2%, 5.6%, 26.9%, and 26.3% of adalimumab-, vedolizumab-, tofacitinib-, and ustekinumab-treated patients, respectively, would achieve induction of remission.

Endoscopic Remission: With an estimated placebo rate of achieving endoscopic improvement of 15% in included trials, we estimated that 16.3%, 17.7%, 45.4%, and 39.1% of adalimumab-, vedolizumab-, tofacitinib-, and ustekinumab-treated patients, respectively, would achieve endoscopic improvement.

Maintenance therapy: All agents have been found to be equally effective for maintenance of remission in a subset of patients who responded to induction therapy (SUCRA, maintenance of clinical remission and endoscopic improvement, golimumab, 0.69 and 0.58; vedolizumab, 0.63 and 0.76; tofacitinib, 0.69 and 0.69; and ustekinumab, 0.47 and 0.46, respectively).

Specific actions taken: *We have added a new citation and updated the text (line 111-112; Page #3). For the convenience of the reviewers, we have copied and pasted that section below*

“Currently, patients are offered inflammation-reducing therapies that have a widely variable ~18-58% response-rate (where response can be either clinical or endoscopic, with a placebo rate of 3-30% during induction therapy (22), 40% of responders become refractory to treatment within one year (23).”

Comment #3. In the BoNE section of the Methods section the authors claim that their approach discretizing variables (gene expression traits) into two states (high or low) and then classifying the relationships between pairs of variables after application of their Boolean algorithm into one of six types. But nothing is said about why their approach has any advantage over other Bayesian network approaches that also discretize input data into any number of states (for example, in one of the papers cited by the authors, ref 27, the network approached developed by Zhu et al. discretizes into 3 states and similar types of inferences made here can be drawn from the edges in such networks as well). A familiar theme at this point, where the authors just haven't done enough to place their approach into the context of what others have done, and then clearly articulate why what they have done offers an advantage. Usually with such methodologic claims comes direct comparisons to alternative

approaches already in use to establish the advantage. If the authors want to make the claim, they should show it.

Response #3: *This comment is specifically about Figure 2 and Figures S2-4 and has to do with head to head comparisons between methodologies. We apologize for the confusion, which appears to have risen from our inability to explain logically what was done, and why.*

We have compared our approach directly with Zhu et al.'s Bayesian approach in Fig 2F. Peters et al., had optimized the Bayesian approach that was developed by Zhu et al., to pick that the best algorithm that serves to analyze IBD datasets. According to their methodology section, Peters et al., constructed the network using the algorithm implemented by RIMBANet software (developed by Zhu et al) and visualized by Cytoscape 3.4. The RIMBANET software for constructing Bayesian networks is freely available (see URLs) and comes complete with instructions on how to run the software and specific examples with step-by-step instructions on reproducing previously published results with this software.

Now, the reviewer is right that the bar for claiming "superiority" is typically higher than "non-inferiority". That is because it is very hard to establish that one method is better than the other. Each method is most likely optimized to ask an important biological question. We first noted that both methods work well in separating healthy and IBD samples (Fig S4A-B). Then, we go on to show that the Boolean method is better than Zhu et al., Bayesian approach in a specific setting where the prediction of outcome is done in a prospective manner; there we show that the Boolean approach performed better in separating Responder vs Non-responders. In this study, the "response" was defined as endoscopic remission ($SES-CD \leq 2$ at week 24 for CD and Mayo endoscopic sub-score ≤ 1 at week 10 for UC), which is one of the outcome measures in any induction therapy study. .

Specific actions taken: *We have clarified this in the revised manuscript (Line 151-161 on Page #5). We also toned down the last sentence to say "may be superior" at this stage, because we felt that declaring superiority at this early stage is unnecessary as well as inappropriate because more head-to-head methodology comparisons are done rigorously later in Fig 5. For the convenience of the reviewers, we have copied and pasted that section below—*

"We compared our approach directly with differential and Bayesian approaches; the latter was optimized by Peters et al. (2) for the analysis of IBD datasets. Despite minimal overlaps between differentially regulated genes across these independent cohorts (Fig S3), conventional approaches e.g., differential and Bayesian performed equally well in separating the healthy and IBD-afflicted samples (Fig S4A-B). However, when it came to distinguishing responders from non-responders in the only prospective study to date where colons were analyzed by RNA-Seq prior to the initiation of treatment with TNF α -neutralizing mAbs [E-MTAB-7604; (3)], Boolean analysis was more accurate than the other two approaches [Figure 2F; ROC-AUC for Boolean, 0.86; Differential, 0.68; Bayesian, 0.61]. These findings indicate that the Boolean approach may be superior in predicting therapeutic response [response was defined as endoscopic remission, (3)]. Additionally, BoNE revealed the ability of the C#1-3 signature to segregate samples according to the aggressiveness of disease consistently across five additional validation cohorts (Figure 2G); it could separate active from inactive disease (4, 5), responders from non-responders

receiving two different biologics, Infliximab (6) or Vedolizumab (7), and even distinguished those with quiescent disease with or without remote neoplasia (8) (Figure 2G). These findings demonstrate the power of Boolean networks in accurately modeling gene expression changes that occur during IBD pathogenesis and predicting clinical outcomes.”

Comment #4. The description of the network construction and identification of BIRs is confusing. Figure 2A makes it appear that the expression are input into a network construction process that delivers Boolean networks and then BIRs are identified from the networks. But the description on page 4 in the methods indicates first step is discretization, second step is classifying relationships between gene pairs, and third building a network by clustering gene pairs. The supp material then tries to clarify that it is the discretization process that delivers the BIRs, which makes Figure 2A appear misleading. The authors need to describe this with enough clarity to enable the results to be replicated.

Response #4: *We agree that the supplementary method is not clear because one of the statements was deleted by mistake. We apologize for this erroneous omission. To be clear, the reviewer is right that the process begins with discretization of data for each gene. BIRs are discovered by evaluating every possible gene pairs using BooleanNet algorithm. Finally, genes are clustered as opposed to ‘gene-pairs’ using Boolean equivalent relationships. Hence, Fig 2A in the original submission and the Methods section within the main text of the original submission was accurate and clear all along.*

Specific actions taken: *We have extensively revised the text in the detailed methods in Supplementary online materials (section Boolean Analysis). The section of the main text with description of methods was accurate (the reviewer’s interpretation of it was also accurate), and hence, no changes were made. The same goes for Fig 2A, which was not changed. For the convenience of the Editors and Reviewers, all these edits are highlighted.*

Comment #5. It would seem critical to fully describe the datasets the authors are using to build their networks since it forms the basis of everything they infer. But the authors only point to references, and it appears they point to incorrect references, so again confusing what was done, and then the data being used, how comparable the data are, what states the samples from which the data were derived were in, and so on, not at all described, so really impossible to assess what they have coming out of those sets.

As an example, on the confusion, the authors point to references 26 and 27 as the source of the datasets. Reference 26 is to a paper by Arijs et al. in Plos One from 2009. Then in the supp the authors point to GSE73661, which is from a 2018 paper by Arijs et al. (ref 32). Does GSE73661 include data from both papers? So they were both used?

Regarding comparability of the two datasets used to build the networks, for the Peters 2017 set, the authors point to GSE83687 and indicate it was comprised of 60 controls, 32 UC, and 42 Crohn’s disease samples? They indicate an n = 143, but it looks like 134 samples? Where does the 143 come from? And in this dataset they are comprised of UC and Crohn’s patients; are the authors then just looking at networks that reflect common parts of those diseases? Are these distinct diseases or the same? Do they occur in the same part of the intestines? If they are in different parts is the assumption that the molecular mechanisms are the same regardless of location? And then the controls were actually not health but rather were colon cancer patients? And given the IBD patients in this cohort were undergoing surgical resections, they seem to represent the most severe forms of IBD not responding to any treatments. This is in contrast to the Arijs 2018 dataset, which seems to have focused only on UC patients and colon biopsies, with perhaps a more appropriate control group (those undergoing colonoscopy, so more healthy, although perhaps older?) The description of GSE73661 is 120 samples from 44 UC patients and then 12 non-IBD samples and 23 more biopsies from UC patients before IFX treatment? And the other 120 were on a different treatment protocol over

time? The authors indicate 178 samples, but again not clear what these all are and without spending a day digging into GSE73661 and GSE83687 datasets (I can't burn days as a reviewer trying to figure out what was done), it is simply not easily possible to figure out what exactly was used here.

There is zero description or discussion from the authors on these different samples and why they could all be combined together, what impact age differences played with controls, what role drug treatments played since Arijs samples were involved in clinical trials, what role severity played given the severity of the Peterson 2017 samples, what role the different intestinal regions played given there was a mix, what role biopsies from inflamed vs. non-inflamed played in the signal, and so on and so forth. In the end, it is difficult to understand what is actually being discriminated here.

Response #5: *In this comment, the reviewer makes three major points:*

- 1) *Insufficient description of the datasets and incorrect references.*
- 2) *Inaccuracy in stating 'n' (total unique patient datasets), which further confuses the identity of the datasets and references.*
- 3) *Unclear if data from various cohorts was combined to analyze.*

We admit that there was a mistake in the reference. We used data from Arijs et al., 2018 (GSE73661).

Also, n = 143 is a typo. It should be Peters et al. 2017 (GSE83687, n = 134), as the reviewer noted.

The description for Arijs et al., 2018 (GSE73661, n = 178) in the GEO page mentions 120 biopsies. However, the paper mentions that 120 biopsies were from UC VDZ group (41 patients), 46 from UC IFX group (23 patients) and 12 control group (so total 178 biopsies).

We used both Arijs et al., 2018 (GSE73661, n = 178) and Peters et al. 2017 (GSE83687, n = 134). However, we did not combine them before analysis. We analyzed them separately.

*Boolean Implication analysis looks for invariant relationships across all the different types of samples regardless of the conditions and treatment protocols. Therefore, it does not distinguish the sample types when discovering Boolean implication relationships. We assume that there are fundamental invariant Boolean implication formulas that are satisfied by every sample regardless of their type (in this context, it is limited to healthy and IBD colonic biopsies including both UC and CD). This means normal, UC and CD samples share the same fundamental relationships. We performed a search over simple Boolean paths in the clustered Boolean implication network (a score computed based on the linear combination of normalized gene expression values) that separated healthy and IBD samples in another cohort GSE6731 (4 N, 5 UC, 7 CD). We tested how the gene signatures distinguish healthy and IBD samples as they are annotated in other independent colon derived datasets. We have also been able to demonstrate that the concept of invariants works not just in IBD samples, but also in the case when analyzed independently as UC and CD only samples (as shown in **Fig S16** and **S17**).*

Specific actions taken:

- 1) *We have corrected the references (line 401-403 in Main Text, Methods).*
- 2) *Fig 1 is revised with n = 134. The supplementary method is revised with GSE83687, n = 134.*
- 3) *We have revised text (line 404-406 on Page #14 in the main MS) and supplementary methods (**section Inflammatory bowel disease (IBD) datasets used for network analysis**) to clarify the points raised by the reviewer. The fact that the test cohorts were analyzed separately (and not combined) has been explicitly stated in the main text on **Page #14**.*

Comment #6. For identification of BIRs, the authors indicate they used BooleanNet statistics and then indicate $S > 2.5$ and $p < 0.1$ as the thresholds, without any description on what these statistics are, how the FDR was computed and so on.

Response #6: *In our original submission we had cited our previous paper on Boolean Implication relationship [D. Sahoo, D. L. Dill, A. J. Gentles, R. Tibshirani, S. K. Plevritis, Boolean implication networks derived from large scale, whole genome microarray datasets. Genome Biol 9, R157 (2008)]. We agree that a citation may be insufficient, and more detailed description here is necessary to help review the methods significantly and allow others to reproduce our work or apply this methodology to other diseases and datasets.*

Specific actions taken: *We have extensively revised our Supplementary Methods to add details of BooleanNet statistics and FDR computation (section Boolean Analysis).*

Comment #7. Regarding the construction of the Boolean implication network, the authors claim on page 3 of supp that “BIR is strong and robust when the sample sizes are usually more than 200.” No reference or support of this claim is given, because as indicated above, the samples being used are so varied, across so many conditions (age, severity, tissue type, disease type, etc.) that it simply can’t be true that some general simulation-based result that indicates BIR are strong with $N > 200$ carries over into this setting. Despite this more varied setting and that the number of samples is < 200 , there is nothing shown that would indicate the results as being “strong and robust”.

Response #7: *We agree with the reviewer that we need to revise our claim here. There is no doubt that higher ‘n’ used in identifying BIRs improves the confidence in the model. All our previous papers used thousands of diverse samples to establish Boolean implication relationships. The threshold of $n = 200$ is more of a ‘set minimum’ based on our experience in building these kinds of networks on diverse datasets in different disease states, rather than formal simulation. But the availability of a single large cohort in some diseases is impossible. IBD was the first time where we are trying to identify Boolean implication relationships in 134 samples which is low, but the nature of the 134 samples made it worth it, for example, this is the only dataset where a ‘full thickness’ colon wall analysis was done in an adult IBD cohort. Such property ensured that transmural processes were captured (not just mucosa). To overcome the low confidence issue, we took two measures: 1) We needed to adjust our BooleanNet statistic threshold ($S > 2.5$ and $p < 0.1$; $FDR < 0.001$) for this. 2) we filtered the analysis through a second cohort [Test cohort #2: Arijs et al., 2018 (GSE73661, $n = 178$)] without combining them at any point. A positive result in our manuscript suggest that Boolean implication can still be carried out with such low number of samples. Ultimately, we have demonstrated that the results are reproducible across many human IBD datasets (see Table 1).*

Specific actions taken: *We have revised supplementary methods, added a new section “Test Cohort Selection”, included a comprehensive new Table (Table 1) which documents not just the size and samples in the independent datasets, but also highlights the heterogeneity of the samples (section Construction of a Network of Boolean Implications). We cite Table 1 in multiple places (text, legends) and also in Supplementary Methods to expand on these details.*

Comment #8. For the clustered BINs, it is not clear if this is a new procedure or if it has been better vetted in other publications. This matters because the authors describe a procedure that conceptually makes sense, but where the selection of thresholds used to actually carry out the clustering seem very arbitrary, with no sensitivity analyses provided that would indicate how the results may vary if different thresholds were used. For example, on page 4 of the supp, the authors describe an approach to ensure internal consistency in the resulting clustered BIN (e.g., avoiding cases in which two genes that are opposite of each other become part of the same connected component) that involves computing the minimal spanning tree for the CBIN, and then computing Jaccard similarity coefficients for all edges, and then rationalizing that if two members share less

than half their total individual connections (Jaccard similarity coeff < 0.5), they should be dropped as they are potentially internally inconsistent. But why 0.5, why not 0.7, or 0.8, or 0.9, or 0.4, etc.?

Response #8: *Although the Boolean logic to studying gene expression relationships is not new (has been around for over a decade; link here), the clustered BINs is a new procedure. This is the first time that this approach is being used and vetted. We hope to address all the concerns by the reviewer in this manuscript. A simple clustering approach made one big cluster with almost all the genes in it because of the noise tolerated by the statistical thresholds. We wanted to break it down to a reasonable number of clusters because one cluster with everything in it will not be useful for downstream analysis. The choice of the threshold on the Jaccard similarity coefficient play an important role in determining the size and the number of clusters as well as whether they are internally consistent. We found that a threshold of 0.5 gave us reasonable number of clusters and followed a power law distribution in the cluster sizes. A bigger threshold such as 0.7 to 0.9 will be very aggressive and reduce the cluster sizes (almost all edges will be dropped). A smaller number such as 0.4 will tend to make bigger cluster with an unusual distribution of cluster sizes.*

Specific actions taken: *We have revised supplementary methods to address this issue in a section entitled “Generation of clustered Boolean Implication Networks”. More specifically, we have included details regarding – 1) how the thresholds were selected; 2) sensitivity analysis and rationalized choice of how to break down larger clusters into smaller clusters while ensuring that these are internally consistent.*

Comment #9. In trying to understand the network results described in the first section of the results, the referenced figures 2 and S1 did not seem to depict a network, but rather provide some schematics (as in Figure 2B) and then some clustering results and then performance metrics, and I could not find Table 1 (there was a Table 1 legend in the supp, but I could not find the table anywhere in the submission – perhaps I missed it, but certainly it was not obviously placed in the main text, the supp, or in the zip file I downloaded from the NCOMM site). So basically, I have no way of actually understanding the result.

Response #9: *We apologize for an extremely lengthy (> 100 page long) single PDF [Supplementary File] which had Tables tucked in, but not accessible, or intuitively locatable.*

Specific actions taken: *To avoid the issue mentioned above, all the network related results are now uploaded as separate file “Tables”. We believe that the final revised draft has everything needed for the reviewer to assess the results.*

Comment #10. In first paragraph of the results the authors claim that using reactome pathway analysis on clusters that emerge from the BIN, reveal the biological processes that the clusters *control*. How can one infer from the clusters that they are actually controlling the biological processes? What is actually meant by that?

Response #10: *We agree our choice of the word “control” was not appropriate. We recognize that reactome pathway analysis of gene clusters can go as far as providing functional associations with biological processes, but not necessarily their ability to impact the process.*

Specific actions taken: *We have amended the results to reflect this change in terminology (line #127 on Page #4). For the convenience of the reviewers, we have copied and pasted that section below—*

*“Reactome pathway analysis of these clusters along the path continuum revealed the most important biological processes **with which they associate** (Figure 2C).”*

Comment #11. What does this network actually reflect? Is it some average over all of the samples that were used to construct the network? The authors indicate in the first paragraph of the results and the supp on “discovery of paths...” that they get a time series of the biological processes starting with healthy through to severe disease. I do not see how this could be possibly by looking at Boolean paths from a network constructed by simultaneous consideration of varied samples. I would agree there may be ways to identify clusters that are reflecting health and disease states given those states are well enough annotated in the samples (you have control and disease individuals), but then for disease progression, you have such varied conditions, different diseases in the case of Peterson dataset (UC and Crohn’s that were characterized using different tissue types), different drug treatments, etc. There is absolutely no way to distinguish between time-based disease progression and differences in states among the different samples used to construct the network. Or if there is, that needs to be very clearly articulated and it simply is not in the supp (Boolean paths are explore which just represent paths through the network and again what exactly the network represents is not really discussed, but certainly it does not naturally represent a time series).

Response #11: In this question the reviewer raises several good points, all of which has to do with missing clarity in how we have described things. More specifically, the reviewer points out that we have not really explained what we mean by Boolean paths reflecting “a time series of biological processes...”.

We agree with the reviewer that the statements are unclear; we should have cited relevant prior work and explained along the way to avoid the perception that the claims are unsupported by logic/data. We have published before an algorithm called MiDReG that identify developmentally regulated genes essentially a time-series of events when asymmetric Boolean implication relationships (in diverse human dataset of cancer and normal tissues) are used using seed genes that mark the end points. We have validated this concept in the context of B cell differentiation (Sahoo et al., PNAS 2010) where Boolean paths represented time series events of gene expression. Here, using MiDReg algorithm/concept we traverse the network to identify paths of clusters where the start and end clusters in the clustered Boolean implication network mark the end points of a possible set of events from healthy to disease. Once the path to be queried is identified, we then ask which end has more healthy or more diseased samples. Based on the orientation of the path (i.e., healthy vs disease end), and the concept that there are time series of events in any biological data, it is hypothesized that our algorithm might identify some of the underlying characteristics of the time series events during disease progression.

The reviewer is right, in that, when working on a cross-sectional biological dataset (such as Peters et al., or Arijis et al cohorts), it is impossible to validate whether we have identified actual time series in the biology of progression.

Specific actions taken: *We have revised our Supplementary methods to include a specific subsection entitled “Charting Boolean Paths” where we have included language to address this issue in the text (line 100, 158-160).*

Comment #12. In second section of results, page 5, the authors talk about distinguishing between IBD and “healthy” using machine learning applied to the gene clusters. But the “healthy” controls are not necessarily healthy given in the case of Peterson they have colon cancer. Further, there is likely severe confounding of the controls given their non-IBD state is likely highly correlated with age (given in the Arijis 2018 controls are undergoing colonoscopy, which is generally only done when you are older). Thus, how do you know you are not picking up signal that reflects cancer given there are well known “field” effects induced by a tumor in normal tissues? How do you know you are not simply picking up an age signature? There are independent validation

datasets that do seem to support the authors having a meaningful classifier, but again nothing is made easy for the reviewers, there is no description of the datasets but rather just GSE identifiers are given. **The authors need to make a table that describes each of the datasets used, what comprised these datasets (tissues profiled, technology used to profile, what diseases were represented, ages, treatment conditions, etc.)**

The authors also don't say much of anything about what the actual machine learning algorithms were that were used to build the classifiers, other than brief mentions of Boolean, Bayesian and differential approaches in Figure 2F). There are hundreds, maybe even thousands of machine learning algorithms, and how these algorithms are trained and then tested matters a great deal in understanding the results. It is impossible as far as I can see in the paper to in any way repeat what the authors did here, since they just mention some methods and give some results of their performance in the figures, without actually describing how the methods were fit, etc. There is a section in the supp that describes "measurement of classification strength or prediction accuracy", but nothing on the different classifiers that were built, apparently there were several to distinguish health from disease, response from non-response, active from inactive disease, etc. Are the 7 validation cohorts used seem to focus just on normal vs. IBD, but then there are claims of predicting these other states as well. It is of note that the precision and F1 scores for the normals are generally not very good, whereas the classification to IBD state seems generally good, although cases where it fails (validation 6), but none of this is discussed.

Response #12: *This is a 2-part comment:*

In the first part, the reviewer asks that we inform the reader of the nature of the healthy control subjects that were used in this study. We agree that these controls are partly from patients undergoing screening colonoscopy, and partly from those undergoing colon resections due to tumor or diverticulitis. Because the choice of controls can impact findings, we have added a table of datasets (Table 1 cited in the main text, Fig 1 legend, Methods in the main text and in Supplementary under the subtitle-- "Test and Validation IBD Datasets") with all the relevant conditions identified by the reviewer. Given that our findings held true across numerous IBD datasets, regardless of the type of controls used, we believe that the findings are unlikely to be confounded by variables that the reviewer mentioned.

We have described the machine learning part in a new section in the supplementary method "AI guided discovery of Boolean paths".

Specific actions taken:

- *Table 1 added with information regarding healthy controls. Cross-cited Table 1 in appropriate places.*
- *We have revised supplementary method to include two new sections titled "AI guided discovery of Boolean paths", and "Test and Validation IBD Datasets".*

Comment #13. For the identification of targets, other than membership in the clusters, the identification was not really carried out in a very network-directed way as promised in the intro and discussion, but rather the authors looked at membership in the C#1-2-3 clusters and matched genes in those to "druggable classes of receptors, enzymes and signal transducers. They identified 17 targets annotated as "response to stress", and then focused on the kinases and in particular the one kinase that had existing specific and potent agonists available, PRKAB1 (AMPK). There was no prioritization based on what the network would have predicted the impact would be on any of the members of these clusters. Do the authors think they would all have the same impact? Is the claim they are all causal? This is not really described.

Response #13: *Drug targets were first filtered through membership in the C#1-2-3 (this is network guided). Our analysis predicted that each of these genes in clusters #1-3 would be good targets. But, as usually is done*

in drug discovery, we prioritized based on 4 commonly used methods: 1) prioritizing 'druggability' of the targets because the vast majority of our proteome may not be druggable; 2) prioritizing availability of potent and specific compounds that minimize the concerns surrounding off-target effects; and 3) sound biological rationale (a plausible MoA); and 4) availability of companion markers.

Applying these criteria, PRKAB1 appeared to be a good target that fulfilled all 4 criteria. This was particularly intriguing to us because PRKAB1 is a subunit of a specific trimeric form of AMPK, a kinase that has been long recognized as a vital player in the maintenance of epithelial polarity and junctions, but never successfully exploited for IBD therapeutics. Similarly, potent PRKAB1 agonists had existed in our therapeutic armamentarium for over a decade, but successful pairing to any indication had not happened.

Specific actions taken: We have described these post-network rationalization process in the text (lines #177-190; pages #5-6). For the convenience of the reviewers, we have copied and pasted that section below

"We next prioritized target genes in C#1-2-3 based on 4 commonly used methods: i) 'druggability' of the targets; ii) availability of potent and specific compounds; iii) sound contextualized biological rationale; and vi) availability of companion markers. To assess 'druggability', gene ontology (GO) molecular function analysis of C#1-3 was carried out, identifying receptors, enzymes and signal transducers that can be targeted easier than other molecules (Figure 3A). Of these druggable interfaces, 17 targets were identified as associated with GO biological function of 'response to stress'. Two of 17 were kinases, of which only one, PRKAB1(β 1 subunit of the metabolic master regulator, AMPK) had commercially available and extensively validated specific and potent agonists with known structural basis (37-39) (Figure 3A). When proteins encoded by C#1-6 were analyzed for cooperativity between cellular processes within protein-protein interaction (PPI) networks using STRING (40), PRKAB1 and other subunits of AMPK appeared at the crossroads between 'pathogen-sensing', 'autophagy' and epithelial 'tight and adherens junctions' and 'polarity complexes' modules (Figure S5).

Comment #14. Regarding the proposed mechanism of PRKAB1 in the gut lining, what the authors proposed on page 7 of the results, appears to already be known? The authors indicate at end of first paragraph on page 8 that their algorithm predicts for the first time that PRKAB1-agonists would "work through upholding epithelial polarity and TJ integrity, which in turn should reduce inflammation." And papers such as Front Pharmacol. 2018 Jul 16;9:761 show the same kind of thing, where this paper indicates that AMPK activation maintains the structure of tight junctions in colonic epithelial cells, reducing inflammation, reducing DSS damage, etc. Not to say that the authors model does not support this exactly, but at least how described, that the identification of PRKAB1 was not totally data driven from the model the authors constructed, but rather met some criteria on what might make a good target and then the story around that was more or less fit to what is known.

Response #14: We agree that AMPK has previously been widely touted by the cited literature as a kinase that augments epithelial barrier. In fact, on page #6 (lines #190-192) we ourselves wrote immediately after prioritizing PRKAB1 as a target—

"This was hardly surprising because AMPK's role in the stabilization of the gut barrier has been known for over a decade (9-11); however, beyond weak agonists like mesalamine (12, 13), nothing has emerged, and the use of relatively more potent but non-specific agonists such as Metformin in IBD is limited due to symptoms of GI intolerance."

The fact that AMPK is a widely recognized target did not dissuade us because we assumed that targeting the predicted PRKAB1 subunit might offer specificity, accuracy and efficacy that Metformin has been unable to give to date.

Hence, on page 6 Line #196 we have now added-- “We hypothesized that our AI-guided approach may have pinpointed a specific subtype of AMPK (i.e., trimers of the kinase that includes PRKAB1) that is important, and that PRKAB1-specific agonists may offer a higher degree of precision and efficacy over non-specific AMPK agonists.”

Specific actions taken: To mitigate the concerns of this reviewer about lack of clarity as to how Clusters 1-2-3 were prioritized to continue with one gene (PRKAB1), we have diligently edited several sections on Page #5-6. For the convenience of the reviewer, we have copy-pasted that section here in the rebuttal document (see below).

“.....“We next prioritized target genes in C#1-2-3 based on 4 commonly used methods: i) ‘druggability’ of the targets; ii) availability of potent and specific compounds; iii) sound contextualized biological rationale; and vi) availability of companion markers. To assess ‘druggability’, gene ontology (GO) molecular function analysis of C#1-3 was carried out, identifying receptors, enzymes and signal transducers that can be targeted easier than other molecules (**Figure 3A**). Of these druggable interfaces, 17 targets were identified as associated with GO biological function of ‘response to stress’. Two of 17 were kinases, of which only one, PRKAB1(β 1 subunit of the metabolic master regulator, AMPK) had commercially available and extensively validated specific and potent agonists with known structural basis (14-16) (**Figure 3A**). When proteins encoded by C#1-6 were analyzed for cooperativity between cellular processes within protein-protein interaction (PPI) networks using STRING (17), PRKAB1 and other subunits of AMPK appeared at the crossroads between ‘pathogen-sensing’, ‘autophagy’ and epithelial ‘tight and adherens junctions’ and ‘polarity complexes’, modules (**Figure S5**). This was hardly surprising because AMPK’s role in the stabilization of the gut barrier has been known for over a decade (9-11); however, beyond weak agonists like mesalamine (12, 13), nothing has emerged, and the use of relatively more potent but non-specific agonists such as Metformin in IBD is limited due to symptoms of GI intolerance.

Proposed mechanism of action of PRKAB1 in the gut lining (page 6 lines 195-209)

We hypothesized that our AI-guided approach may have pinpointed a specific subtype of AMPK (i.e., trimers of the kinase that includes PRKAB1) that is important, and that PRKAB1-specific agonists may offer a higher degree of precision and efficacy over non-specific AMPK agonists. Mechanistically, they may augment epithelial tight junctions (TJs) in the presence of pathogens by activating a specialized signaling program in the epithelium lining the gut, the stress polarity signaling (SPS) pathway (18) (**Figure 3B**). The SPS pathway involves the phosphorylation of the polarity scaffold, Girdin (GRDN) at a single site (Ser245) by AMPK, an event that appears to be both necessary and sufficient for the strengthening of epithelial junctions under bioenergetic stress. Because the SPS-pathway is triggered exclusively as a stress response, and improves modular cooperativity within the PPI network, it fulfills the criteria of “creative elements” (19); the latter are believed to be critical for the evolvability of complex systems and their pharmacological modulation is predicted to help survive unprecedented challenges/stressors. More importantly, in a related manuscript (20), we confirmed using PRKAB1-specific agonists that the SPS-pathway as a putative cell-type specific ‘companion’ biomarker for AMPK activation in the gut epithelium.”

Comment #15. Given the authors focus on a target that is already somewhat well known and in particular with respect to the IBD hypotheses derived by this manuscript, why not apply this approach more objectively to identification of all targets, the hypotheses relating to mechanism that obtain given the models and

annotations, and then assess both positive and negative accuracy with respect to appropriately recapitulating what is known about current therapeutics that worked versus those that failed. The authors present something pretty intriguing along these lines as detailed in figure 5, but then this was only done in a “guilt-by-association” way, where if connectivity to ampk was shown, then positive, else negative. I think one of the more interesting aspects of the paper. Question is whether other of the targets for FDA approved drugs would have been similarly uncovered like PRKAB1 and then in turn implicated the others that were FDA approved, whether the biology uncovered was consistent with what was known, etc. I think this would make the result far more powerful.

Response #15: We agree. In this revised submission, we have now performed the analysis suggested by this reviewer. We used all of the targets that are either approved or abandoned and performed Boolean association (equivalent) with each other to see if the FDA approved targets tend to implicate each other. We used equivalent instead of hi/lo or opposite because these targets are all anti-inflammatory so they should be positively associated with each other. We saw that the FDA approved targets overwhelmingly implicate each other. Most of the abandoned targets do not implicate any of the approved targets. There is a small number of abandoned targets that are associated with approved targets. We thank the reviewer for this extremely clever suggestion for an analysis.

Besides this, we should also point out that **Figure S18** in the original submission was dedicated to showcase how failed and successful targets were either found or missing from the prominent Boolean paths that were identified in this study.

Specific actions taken: A new panel C is added to **Figure S19** that shows which targets are associated with the FDA approved targets. The Figure legend and main text (**Page #12; lines #375-380**) have been modified accordingly.

C. Boolean Equivalent relationships (in GSE83687) between FDA approved and abandoned targets. FDA approved targets tend to implicate each other using Boolean Equivalent relationship.

For the convenience of the reviewer, we have copy-pasted that section in the rebuttal document (see below).

“BoNE successfully distinguished the FDA-approved vs. the abandoned targets (ROC AUC 1.00; Accuracy 1.00; Figure 5D, S19). By contrast, all targets were significant by differential analysis (high false positive rate; Figure 5D, S19) and almost all the ‘successes’ were missed by Bayesian analysis (high false negative rate; Figure 5D, S19A-B). Furthermore, a Boolean association analysis among targets (FDA-approved and failed) was carried out to see if targets tend to implicate each other. We used equivalent instead of hi/lo or opposite because these targets are all anti-inflammatory so they should be positively associated with each other. FDA approved targets were found to overwhelmingly implicate each other (Figure S19C), whereas most of the abandoned targets do not implicate any of the approved targets. Findings indicate that BoNE can accurately assess the probability of a target to pass efficacy test in Phase III clinical trials.”

Comment #16. Regarding the network-rationalized murine model selection process, the question would be if you did not use the network model at all, but instead took a look at what is known today on ampk, processes relating to IBD it is involved in, known role agonists play in those processes, would you have come up with the same mouse model? Or would that have led you astray and the network would have corrected the thinking?

Response #16: *This is a very important question, one that asks how much does the network influence our pre-clinical model selection, and if we would have chosen the same model regardless of network-based prioritization simply based on the current literature on AMPK and AMPK-driven processes in the gut lining.*

In this regard, we should clarify that AMPK agonist studies have mostly been carried out using Metformin or AICAR, neither likely to translate to the clinic at doses that are effective in animal models. Second, AMPK’s (or its agonists) role has been demonstrated in several murine models of colitis, including DSS(12), TNBS(21-23), IL10-/(24, 25) and Adoptive T-cell transfer models(26). Despite these decade old evidence, in multiple models, targeting AMPK in the gut barrier had not been realized. Our model suggested that many of these models were appropriate (DSS, TNBS, adoptive T cell transfer), whereas some others were not (IL10-/-). We admit that conventional thinking would have also most likely led us to choose either DSS-induced or adoptive T cell transfer colitis as models, and either would have given us comparable efficacy.

However, during the time it took us to write and revise this work, we took another set of targets directly out of the network and validated their use as first-in-class macrophage modulators that are important for protecting the gut barrier. In this instance, the IBD-map indeed helped prioritize the pre-clinical model of colitis most suited to test that particular target class. Briefly, we used the same IBD network (presented in the current work) and created a user-friendly interface to query any target gene and create what we call-- a ‘target report card’. Such report card then helps navigate each and every step in drug discovery. We have presented here the Title, Abstract and Figure 1 (Target report card) sections of the related manuscript. What is important for us to show is that in this second example, the conventional models used by others in this target class (largely, DSS) was not the optimal (low score). Better models existed, which were used by us to pinpoint macrophage-mediated bacterial clearance as the core mechanism of action. So, while we cannot say that network-driven pre-clinical model prioritization would have differed in the instance of PRKAB1, we are certain that it did differ in the case of the next target/drug.

AI-guided use of balanced PPAR α / γ -dual agonist finetunes macrophage responses in inflammatory bowel disease (13 words)

Authors: Gajanan D. Katkar¹, Vanessa Castillo¹, Mahitha Shree Anandachar², Vidales, Eleadah¹, Ibrahim Sayed^{2S}, Daniel Toobian¹, Fatima Usmani², Sahar Taheri³, Dharanidhar Dang³, Joseph Sawires⁶, Jerry Yang⁶, William J. Sandborn^{7*}, Soumita Das^{2*}, Debashis Sahoo^{3-5*} and Pradipta Ghosh^{1, 5, 7, 8*}

Abstract (250 words)

A computational platform, the Boolean network explorer (*BoNE*), has recently been developed to infuse AI-enhanced precision into drug discovery; it enables querying and navigating invariant Boolean Implication Networks of disease maps for prioritizing high-value targets. Here we used *BoNE* to query an Inflammatory Bowel Disease (IBD)-map and prioritize two nuclear receptors, PPAR α / γ . Balanced agonism of PPAR α / γ was predicted to impact macrophage processes, ameliorate colitis in network-prioritized animal models, ‘reset’ the gene expression network from disease to health, and achieve a favorable therapeutic index that tracked other FDA-approved targets. Predictions were validated using a balanced and potent PPAR α / γ -dual agonist (PAR-5359) in two pre-clinical murine models, i.e., *Citrobacter rodentium*-induced infectious colitis and DSS-induced colitis. Mechanistically, we show that balanced dual agonists, but not individual agonists, promote bacterial clearance more efficiently than alone both *in vivo* and *in vitro*, through the controlled induction of pro-inflammatory cytokines and cellular ROS. PPAR α is required and its agonism is sufficient to induce the pro-inflammatory response that is essential for bacterial clearance and immunity, but PPAR γ -agonism blunts these responses, delays clearance and induces the anti-inflammatory cytokine, IL10. Balanced agonists achieved controlled inflammation and barrier protection. When tested on PBMCs derived from 10 patients with Crohn’s Disease (CD), PAR-5357 reversed the defective bacterial clearance observed in these subjects. Findings not only deliver a macrophage modulator in IBD but also highlight the potential of *BoNE* to accelerate and enhance the precision of drug discovery in various diseases.

Figure 1: Network-rationalized target identification and study design. (A) Schematic displays the overall computationally guided study design. An interactive web-based platform allows the querying of paths of gene clusters in the IBD map (Sahoo D., et al., Under Review) to pick high-value targets with a few mouse clicks and generate a comprehensive automated target ‘report card’. The components of a ‘target report card’ is shown (*right*): predicted ‘therapeutic index’ (likelihood of Phase III success), IBD outcome (prognostic potential in UC and/or CD), network-prioritized mouse model, estimation of gender bias and predicted tissue cell type of

action. **(B-H)** Components of a target report card for PPARA and PPARG are displayed. Bar plot (B; top) displays the rank ordering of normal vs ulcerative colitis (UC) /Crohn's Disease (CD) patient samples using the average gene expression patterns of the two genes: PPARG/PPARA. ROC-AUC statistics were measured for determining classification strength of normal vs IBD. Bar plots (B; top) and violin plots (B; bottom) display the differences in average expression of the two genes in normal, UC and CD samples in the test cohort used to build the IBD-map. Bar plots in panel C-D show the rank ordering of either normal vs IBD samples (C) or responder vs non-responder (R vs. NR; D), or active vs inactive disease, or neoplastic progression in quiescent UC (qUC vs. nUC; D) across numerous cohorts based on gene expression patterns of PPARG and PPARA, from high to low, left to right. Classification strength within each cohort is measured using ROC-AUC analyses. Bar plots in panel E show the rank ordering of either normal vs IBD samples across numerous published murine models of IBD based on gene expression patterns of PPARG and PPARA as in D. ACT = adoptive T cell transfer. Classification strength within each cohort is measured using ROC-AUC analyses. Bulk = whole distal colon; epithelium = sorted epithelial cells. Schematic in F summarizes the computational prediction of cell type of action for potential PPARA/G targeted therapy, as determined using Boolean implication analysis. GSEID# of multiple publicly available databases of the different cell types and colorectal datasets used to make sure predictions are cited. Red boxes/circles denote that PPARA/G-targeted therapeutics are predicted to work on monocytes/macrophages and crypt-top enterocytes. Computationally generated therapeutic index (see *Methods*) is represented as a line graph in G. The annotated numbers represent Boolean implication statistics. PPARA and PPARG align with FDA approved targets on the right of threshold (0.1). Two FDA approved targets (green; ITGB1, 0.046; JAK2, 0.032), two abandoned targets (red; SMAD7, 0.33; IL11, 0.16), PPARA (grey, 0.064), PPARG (grey, 0.04), and the threshold (black, 0.1) are shown in the scale. Box plot in panel H shows that the levels of PPARA/G expression is similar in the colons of both genders in health and in IBD, and hence, PPARA/G-targeted therapeutics are predicted to have little/no gender predilection.

Specific actions taken: *We have now informed the reader that of the numerous models previously used to study the role of AMPK in colitis/gut epithelial barrier, DSS-induced colitis was the highest ranked and hence chosen as a preferred model. We agree that this was an important point that we missed emphasizing in the original submission. See lines #275-282 on Pages #8-9. For the convenience of the reviewers, we have copied and pasted that section below*

"We used BoNE to prioritize the murine models of colitis that most accurately recapitulates the barrier-defect transcript signature in human IBD, i.e., downregulation of genes in C#1-3 (Figure 4A). DSS-induced colitis, which triggers intestinal inflammation by compromising the integrity of the gut barrier (59) emerged as the best (for both bulk colon and sorted epithelial cell-derived datasets), closely followed by TNBS, adoptive T-cell transfer and Citrobacter-induced colitis, whereas genetic models were deemed inferior (Figure 4B).

REVIEWER #2:

General Comments:

This is a well **performed extensive study** that uses computational methods to identify new targets to treat IBD. This **approach potentially identifies prediction models to be used at early stages in the clinical pipeline**. In addition, the methods described are used to identify the best preclinical models to test the validity of the target. Overall the **study is well designed**, and the **manuscript well written**.

I have a few comments however I would like the authors to address/respond to.

Response to general comments: *We are very appreciative of the overall positive nature of the comments and the generous praise expressed by this reviewer. We hope that through additional experiments we have adequately answered the few concerns that this reviewer raised. We hope that our answers and the revisions we made in the manuscript mitigate all concerns.*

Comment #1: The training and testing cohorts include quite heterogenous populations. For instance, GSE83687 contains samples from different locations (small bowel including but not exclusively ileum, rectum, sigmoid colon, and right colon). Moreover, both Crohn's disease and ulcerative colitis samples are analyzed together. How did the authors correct for these variables (i.e. disease location, disease type)? Given the marked differences in the transcriptomes of different intestinal locations, transcriptional analysis of these type of samples is normally performed separately. However, in the case of network analysis it is not clear from the manuscript how this was accounted for and how it could have affected the results. Also, while UC and CD share some mechanisms, epithelial dysfunction for instance is one of the main differential pathways between the two diseases. I would predict that had the authors analyzed UC and CD samples separately, the results of their Boolean analysis may have been somehow different. Did the authors perform any analysis on more homogeneous groups of samples (i.e. colonic UC only, colonic CD only or SB CD only).

Response #1: Boolean implication works well when there is heterogeneity. Boolean analysis pulls out invariant relationships that are satisfied by all kinds of tissue. Focusing on homogeneous groups attracts cohort specific expression patterns are not likely reproducible in other independent cohorts. However, it is still possible to think about colonic UC only, colonic CD only or SB CD only as pointed out by the reviewers. We have analyzed colonic UC and colonic CD only samples in **Fig S16** and **S17**.

Specific actions taken: We have added a table (**Table 1**) with the demographics of each cohorts because it was important to highlight the degree of heterogeneity in each dataset. These further go on to show how the gene clusters are invariably altered in one way (downregulation of genes in clusters 1-2-3 and upregulation of genes in clusters 4-5-6) in every sample, regardless of the noisy variables. The fact that the test cohorts (that were used to build the network) were analyzed separately (and not combined) has been explicitly stated in the main text on **Page #14**.

Comment #2: The findings on the superiority of the Boolean analysis to predict response to different treatments (Fig 2F-G) are interesting. Was the C1-3 signature the only one that consistently segregate active from inactive as well as R to NR to both IFX and vedo? How do the authors explain this consistency? Could this prediction of response be related to the severity of the disease (more severe patients tend to respond worse to treatment)? The fact that response to two drugs that target independent MOA can be predicted by the same signature that is also the same that segregates by disease activity would suggest that. It would be interesting though to hear what the authors think about that.

Response #2: Machine learning approach pinpointed C#1-2-3 as the best clusters that consistently separated healthy from IBD samples (**Fig 2E**). These clusters were also the ones that best and most consistently separated H vs IBD samples in independent datasets and worked superior to DEA or Bayesian approach signatures (Fig 2F-G). As to how we explain "this consistency", it is important that we emphasize again that unlike DEA or Bayesian approaches, the Boolean approach relies on 'invariant' relationships that is fulfilled by *all* samples in a dataset. The resultant signature identified a fundamental invariant continuum of disease states from healthy to IBD. Therefore, it is able to separate inactive from active, and responders (R) from non-responders (NR). We think that the response to the two different drugs is related to severity of the disease.

Finally, as showcased here in Pages #23-25 of this rebuttal document, we have begun the process of querying this network for actionable targets with a predicted high therapeutic index with high likelihood of success (FDA-approval) based on how well the targets associate with other successes (see Response to Comment #15 of Reviewer #1, Page #22, where we have displayed a newly added **Fig S19C**).

Specific actions taken: Rewrote "Methods" section in main text elaborating the "invariant" concept (Page #14):

"The other advantage of using BIRs is that they are robust to the noise of sample heterogeneity (i.e., healthy, diseased, genotypic, phenotypic, ethnic, interventions, disease severity) and every sample follows the same mathematical equation, and hence is likely to be reproducible in independent validation"

datasets. *The heterogeneity of samples in each of the datasets used in this study is highlighted in Table 1.*

Comment #3: The authors state in page 8 that “the algorithm is unable to pick the direction, start and end of these events. Our analysis...barrier disruption”. Please explain how this is not in contradiction of Figure 2 C.

Response #3: *The Boolean implication network cannot predict the direction, start and end of the events because it is not aware of the sample annotation i.e., healthy vs IBD. We orient the network direction from healthy to IBD after we identify the nature of the samples and their rank order of distribution. The overwhelming clustering of samples of healthy kind in one end and IBD on the other helps us assign directions.*

Specific actions taken: *We have revised the text to mention that the Boolean implication network is unable to pick the direction, start and end of these events unless the identity of the samples is revealed. See Page 7; Lines #231-234). For the convenience of the reviewer, we have copied and pasted that section below*

“Although the algorithm tries to uncover a timeseries component of the IBD events, the algorithm is unable to pick the direction, start and end of these events; the network direction is oriented later by revealing the identity of the sample types that overwhelmingly cluster at one end vs. the other”.

Comment #4: Could the authors comment on the findings on Supl. Figure 6 (cohort 2), where PRKAB1 expression was maintained lower than healthy mucosa in remitting patients? This seems to contradict their statement (page 8) that “low levels of PRKAB1 correlates with a higher degree of leakiness of the epithelial barrier...”. This correlation is true in active disease, but it appears that inactive disease maintains low levels of PRKAB1 expression in the absence of higher CLDN2 or proinflammatory cytokine expression. Could the authors hypothesize what the implications of this persistent low PRKAB1 expression could be on disease progression. Would the use of specific activators of PRKAB1 could also benefit maintenance of remission?

Response #4: *Yes, we do believe that agonists of PRKAB1 would be effective not just as an adjuvant therapy in acute exacerbation (alongside mainstay anti-inflammatory therapies, with whom the PRKAB1 agonists are predicted to synergize) but also as standalone regimen in maintaining remission. If used in the latter mode, its use is predicted to reduce the other long-term sequelae of IBD, namely, the risk for developing neoplastic disease. The fact that these agonists protected gut epithelial barrier in mice and in organoid models derived from healthy and IBD-afflicted patients (in the setting of LPS or live microbe challenge) tells us that the therapy may prove to be a valuable agent to maintain remission in these patients.*

Specific actions taken: *Because we did not conduct studies or generate evidence to be able to claim the efficacy of its use as an agent to maintain remission, we did not want to go beyond the evidence to make such a claim. We hope that the reviewer will understand this limitation.*

Comment #5: Please revise the images chosen in Figure S9 for active (mod to severe) UC. Specially the second mod to severe UC sample (CLDN2 staining) shows typical architectural crypt disruption commonly see in involved UC samples, but lacks signs of acute disease (epithelial loss, and lamina propria increased infiltrate).

Response #5: *We agree.*

Specific actions taken: *In Fig S9A, the IHC panel for mod-severe UC (CLDN2 staining) has now been replaced with another field that shows not just infiltrates, but also epithelial erosions.*

Comment #6. TEER measurements should be provided at different points over time for each culture, as they

are severely influenced by the number of cells plated, their proliferating capacity and the plate confluency. Indeed, the authors mention in the suppl methods that resistance was measured at 1h intervals for 8 h. How was the time point for Figures 4 I chosen? Also, what was the n for those experiments? Please state in the figure legend.

Response #6: *This question has many parts. The first point accurately points out that TEER is highly influenced by cell #, proliferating abilities, and confluency. We agree.*

We had three different types of monolayer studies in Fig 4:

1) *Steady-state: Fig 4I-J*

The TEER in Fig 4J was measured after monolayers were stably formed and reached a plateau of TEER over two consecutive time points. This typically occurs at 48 h after plating monolayers. The total n for 4J is n = 3-5 reads/patient-derived organoid lines in 4I.

2) *Drug Rx: Fig 4K-P*

When the drug was used to repair the defective TJs in IBD-derived EDMs, TEER was monitored at 4h, 8 h, 12 h and 24 h. The IF panels in Figure 4K-L were after overnight treatment with PF compound (16 h).

We performed pretreatment for 16 hrs for the following reasons

a) At the time of EDM preparation, all the cells were Lgr+ stem cells and they differentiated and formed the polarized monolayers. TEER values at the time of EDM preparations were very low, indicating no differentiated epithelial cells were formed. After 24 hr of EDM preparation, we could see the TEER value in the EDMs is increased. The plateau of TEER values reached within 32 hrs. We also found the Lgr5+ cell population decreases, and the cells form the monolayers within that time period (shown previously, PMID: 32041849 and 32003126). That's why we have selected the pretreatment for 16 hrs, which begins 32 hrs after the preparation of EDMs.

b) Importantly, we always count and seed the same number of cells (2×10^5 cell/ well) for healthy, UC, and CD. For each cell type, either healthy or diseased, we used the same media, transwell and cell number. All the EDMs are performed at the same time to confirm that the cells from different groups have the same conditions.

3) *LPS/microbe challenge +/- drug: Fig 4Q-R*

Here, TEER was assessed every 1 h and the data is shown for end of experiment (which is 8h).

We performed infection with LF82 for 8 hrs since we found that the drop in TEER (invasiveness of bacteria) is started at 8 hrs of infection. We noticed a slight increase in TEER after 2-4 hr of infection, probably because of the compensatory host mechanism to the infection.

Specific actions taken: *The n is now added to Fig 4J. We have also defined the chosen time points for TEER measurement in the legend for Fig 4J.*

Comment #7: *Where were biopsies taken from to generate organoids (active disease? Involved areas?) In figure 5A the effect of other variables such as endoscopic disease activity, time since diagnosis, treatment at*

the time of endoscopy or intestinal segment from which biopsies are obtained to generate organoids, needs to be tested.

Response #8: *We have now added all these additional metadata and repeated the analysis in Fig 5A.*

Specific actions taken: *We have modified the table of patient metadata to add the diseased area, disease score, time since diagnosis, treatment at the time of endoscopy. Figure 5A has been replaced with the new figure where we have added all these above criteria. Our findings remained unchanged, i.e., the change in TEER after treatment with PRKAB1 agonist was not an effect that was confounded by any of these additional variables.*

Comment #8: Please describe how is “aggressiveness of disease” (page 6) defined.

Response #8: *This comment is about a statement that currently appears on page 5 of this revised manuscript. Lines #161-166. As stated in the original sentence (which is copied and pasted below), by aggressiveness we meant treatment response and progression to neoplasia.*

“BoNE revealed the ability of the C#1-3 signature to segregate samples according to the aggressiveness of disease consistently across five additional validation cohorts (Figure 2G); it could separate active from inactive disease (4, 5), responders from non-responders receiving two different biologics, Infliximab (6) or Vedolizumab (7), and even distinguished those with quiescent disease with or without remote neoplasia (8) (Figure 2G).”

Specific actions taken: *None*

Comment #9: The statement on page 11 “combination of....therapeutic signaling” is not clear what it is based on. To my understanding the authors do not provide any data to back that up. That would definitively be a possibility worth addressing though.

Response #9: *We agree that brevity reduced the clarity of the sentence in this case.*

Specific actions taken: *We have now expanded the thought, and the highlighted part has been added during revision, on Page #11, Lines 354-357. For the convenience of the reviewer, we have copied and pasted the section below:*

“Furthermore, combination of PRKAB1-agonists with anti-inflammatory agents is likely to show therapeutic synergy, because these agonists seek to upregulate gene clusters on the healthy side of the network whereas all other FDA-approved agents seek to suppress the expression of pro-inflammatory genes on the diseased side (Fig S18).”

REVIEWER #3:

The authors used an artificial intelligence (AI)-derived process to discover a potential new therapy for inflammatory bowel disease (IBD). The **study is large and addresses the biological complexity of IBD using Boolean modeling**. However, a key challenge in Boolean modeling is to match the right model with the right questions in a specific research context. **The work is novel and rich in data**, but a substantial number of concerns are identified and described below:

1. The authors used and integrated data from two very different and heterogeneous datasets derived from Arijs et al. and Peters et al. published studies. This analysis has the following issues:

- a. Arijs et al. RNA dataset is derived from Affymetrix microarray analysis while Peters et al. dataset is derived from RNA-sequencing analysis.
- b. Arijs et al. RNA samples were extracted from mucosal biopsies, while Peters et al. were extracted from surgical specimens.
- c. Arijs et al. control samples derived from healthy individuals, while Peters et al. derived from colon cancer patients (10cm away from tumor site).
- d. Arijs et al. dataset derives from patients who are refractory to corticosteroids and/or immunosuppression, while Peters et al. dataset derives from patients treated with different treatments.
- e. The authors did not evaluate these clinical differences into their datasets and integrated all gene expression data.

Response #1: *This is an important question that was asked also by **Reviewer #1, Comment #5** (see Page #15) and **Reviewer #2, Comment #1** (see Page #26 of this rebuttal document). As emphasized earlier, Boolean implication works well when there is heterogeneity. Boolean analysis pulls out invariant relationships that are satisfied by all kinds of tissue. Focusing on homogeneous groups attracts cohort specific expression patterns are not likely reproducible in other independent cohorts. However, it is still possible to think about colonic UC only, colonic CD only or SB CD only as pointed out by the reviewers. We have analyzed colonic UC and colonic CD only samples in **Fig S16** and **S17**.*

Each dataset was independently analyzed. We tried to identify general properties of the gene expression relationship that are similar across all of the heterogeneity.

Specific actions taken: *In this revised version, the supplementary methods (detailed) section has been vastly expanded to include subtitles that guide a reader who is trying to understand what was exactly done and how to reproduce the work. In addition, a new Table of cohorts and samples across all Test and Validation datasets has been added in the methods section. The fact that the test cohorts were analyzed separately (and not combined) has been explicitly stated in the main text on **Page #14**. For the convenience of the reviewer, we have copied and pasted that section below*

*“Second, gene expression relationships between pairs of genes were classified into one-of-six possible BIRs, two symmetric and four asymmetric, and expressed as Boolean implication statements (Figure 2A). **This offers a distinct advantage from conventional computational methods (Bayesian, Differential, etc.) that rely exclusively on symmetric linear relationships in networks. The other advantage of using BIRs is that they are robust to the noise of sample heterogeneity (i.e., healthy, diseased, genotypic, phenotypic, ethnic, interventions, disease severity) and every sample follows the same mathematical equation, and hence is likely to be reproducible in independent validation datasets**”.*

Comment #2. The authors present in figure 2C, a map of longitudinal gene network analysis from healthy to IBD. This graph shows that tight junction network is the one deregulated early in the IBD development process followed by bioenergetics changes and then immune-related events. This analysis is only appropriate and valid if RNA datasets derived from patient biopsy samples collected in different time points during IBD development, treatment and remission in the same patient samples.

Response #2: *We have developed a hypothesis of healthy to disease progression by using large heterogeneous cross-sectional datasets. The progression is purely a logical derivative, one in which we explicitly state that the continuum states can go bidirectionally and that the map does not know (or claim with*

any degree of certainty) which came first. In the absence of large time series datasets (which do not exist in IBD) with serial biopsies from numerous patients, we are unable to confirm if what the map predicts is the case.

Specific actions taken: We have clarified these points in methods and have explicitly stated this in main text (lines # 231-241; Page 7). For the convenience of the reviewer, we have copied and pasted that text below:

“Although the algorithm tries to uncover a timeseries component of the IBD events, the algorithm is unable to pick the direction, start and end of these events; the network direction is oriented later by revealing the identity of the sample types that overwhelmingly cluster at one end vs. the other. Our analysis simply shows what is a common knowledge in IBD, i.e., if the barrier is disrupted, then it can be permissive to inflammation; the reverse is also true that if there is inflammation, that can lead to barrier disruption. Therefore, a logical interpretation of the Boolean paths is that the state of no inflammation and intact mucosa is both the start point of the disease and the desirable end point of therapeutic goals. The algorithm for the first time precisely lists actionable genes/targets who may help achieve that goal; in this case, PRKAB1-agonists were predicted to work through upholding epithelial polarity and TJ integrity, which in turn should reduce inflammation.”

Comment #3. In Figure 3A, the authors show that 336 druggable gene targets and 62 genes related to responses to stimuli. A subset (n=17) of these 62 genes are related to the stress response. How the authors decided to focus on these 17 stress-related genes. Protein-protein network analysis identified that this 17-gene signature is a key regulator of all these genes? Also, how the authors focused their interest on PRKAB1 and not in the other 16 gene targets, what is their rationale?

Response #3: We apologize that the strategy for target prioritization *after* the unbiased identification of C#1-2-3 genes was not clear outlined. Hence, both Reviewer#1 (see Response #13, on Page #19) and this reviewer accurately pointed out this missing information.

Here is what was used and why: Drug targets were first filtered through membership in the C#1-2-3 (this is network guided). Our analysis predicted that each of these genes in clusters #1-3 would be good targets. But, as usually is done in drug discovery, we prioritized based on 4 commonly used methods: 1) prioritizing ‘druggability’ of the targets because the vast majority of our proteome may not be druggable; 2) prioritizing availability of potent and specific compounds that minimize the concerns surrounding off-target effects; and 3) sound biological rationale (a plausible MoA); and 4) when possible, availability of companion markers.

Applying these criteria, PRKAB1 appeared to be a good target that fulfilled all 4 criteria (and we narrate in the next how each criterion was met). This was particularly intriguing to us because PRKAB1 is a subunit of a specific trimeric form of AMPK, a kinase that has been long recognized as a vital player in the maintenance of epithelial polarity and junctions, but never successfully exploited for IBD therapeutics. Similarly, potent PRKAB1 agonists had existed in our therapeutic armamentarium for over a decade, but successful pairing to any indication had not happened.

Specific actions taken: We have described these post-network rationalization process in the text (lines #177-190; pages #5-6). For the convenience of the reviewers, we have copied and pasted that section below

“We next prioritized target genes in C#1-2-3 based on 4 commonly used methods: i) ‘druggability’ of the targets; ii) availability of potent and specific compounds; iii) sound contextualized biological rationale; and vi) availability of companion markers. To assess ‘druggability’, gene ontology (GO) molecular function analysis of C#1-3 was carried out, identifying receptors, enzymes and signal transducers that can be targeted easier than other molecules (Figure 3A). Of these druggable interfaces, 17 targets were identified as associated with GO biological function of ‘response to stress’. Two of 17 were kinases, of which only one, PRKAB1(β1 subunit of the metabolic master

regulator, AMPK) had commercially available and extensively validated specific and potent agonists with known structural basis (37-39) (Figure 3A). When proteins encoded by C#1-6 were analyzed for cooperativity between cellular processes within protein-protein interaction (PPI) networks using STRING (40), PRKAB1 and other subunits of AMPK appeared at the crossroads between 'pathogen-sensing', 'autophagy' and epithelial 'tight and adherens junctions' and 'polarity complexes', modules (Figure S5).

Comment #4. The authors show that PRKAB1 is a novel target and its mRNA levels in colonic biopsies are decreased relative to controls. First, the number of samples (n=9) is very low to justify the identification of a new clinically-relevant drug target. Second, the authors do not show PRKAB1 mRNA levels in UC relative to CD colonic biopsies and how its expression correlates with different disease location (ileum vs colon) and drug treatment. There is extensive literature showing that IBD disease location is a key factor affecting gene expression patterns.

Response #4: We appreciate the reviewer's comment and the suggestions to add the UC subjects with the difference in their disease location. We are fortunate to have all the requested samples in our biobank and we performed new experiments where we have 8 healthy, 16 UC and 14 CD subjects. We have added UC samples from the left colon, right colon and rectum. We have added CD samples from the ileum and colon.

The new data of PRKAB1 and Claudin2 has been added in **Figure 3E**. We understand the reviewer's concern that the location of disease (i.e., segment of the gut) is a key factor that can affect gene expression patterns. We noticed a part of the CD ileum has higher Claudin2, possibly because of their disrupted barrier. A recent ongoing study from our group indicated that from different CD subtypes (stricturing, penetrating and non-stricturing and non-penetrating, NSNP), the NSNP group of subtypes had lowered TEER and defective barrier compared to the other sub-disease types.

Specific actions taken: We have now generated new data during revised submission. We have assessed PRKAB1 and Claudin2 levels in UC samples from the left colon, right colon and rectum and CD samples from the ileum and colon. We were fortunate to have the wherewithal to be able to generate these data despite a pandemic and clinic closures because of strong translational partnership with UCSD IBD clinic and existing infrastructure and IRB provisions to have allowed these new patient recruitments and analyses. **Figure 3E** has been replaced with these new data. We believe that the original claims and reported findings are now backed up by stronger evidence. Overall, this improves the rigor of the work, and we are grateful for the reviewer to have suggested this.

Comment #5. The identification of the PRKAB1/AMPK pathway in IBD is not novel; there is previous literature related to the role of this pathway in IBD development and intestinal epithelial permeability. In 2010, Bai et al. showed that activation of this pathway by using an AMPK agonist suppresses the inflammatory response in a TNBS-induced colitis mouse model (Bai et al. *Biochem Pharmacol.* 2010 Dec 1;80(11):1708-17). Sun X et al. showed that AMPK activation enhances intestinal epithelial barrier function (Sun X et al. *Cell Death Differ.* 2017 May;24(5):819-831). Furthermore, Chen L. et al. shows that AMPK activation promotes the expression and assembly of tight junctions (Chen L et al. *Front Pharmacol.* 2018 Jul 16;9:761).

Response #5: We agree that AMPK has previously been widely touted by the cited literature as a kinase that augments epithelial barrier. In fact, on page #6 (lines #190-192) we ourselves wrote immediately after prioritizing PRKAB1 as a target—

"This was hardly surprising because AMPK's role in the stabilization of the gut barrier has been known for over a decade (9-11); however, beyond weak agonists like mesalamine (12, 13),

nothing has emerged, and the use of relatively more potent but non-specific agonists such as Metformin in IBD is limited due to symptoms of GI intolerance.”.

Specific actions taken: To ensure that proper credit goes to those before us who have demonstrated the role of AMPK in colitis models, in this revised version of the manuscript the text has been updated in the following ways.

The fact that AMPK is a widely recognized target did not dissuade us because we assumed that targeting the predicted PRKAB1 subunit might offer specificity, accuracy and efficacy that Metformin has been unable to give to date.

Hence, in page 6 Line #196 we wrote-- “We hypothesized that our AI-guided approach may have pinpointed a specific subtype of AMPK (i.e., trimers of the kinase that includes PRKA β 1) that is important, and that PRKAB1-specific agonists may offer a higher degree of precision and efficacy over non-specific AMPK agonists.”

How Clusters 1-2-3 were prioritized to continue with one gene (PRKAB1), we have diligently edited several sections on Page #5-6. For the convenience of the reviewer, we have copy-pasted that section here in the rebuttal document (see below).

“.....“We next prioritized target genes in C#1-2-3 based on 4 commonly used methods: i) ‘druggability’ of the targets; ii) availability of potent and specific compounds; iii) sound contextualized biological rationale; and vi) availability of companion markers. To assess ‘druggability’, gene ontology (GO) molecular function analysis of C#1-3 was carried out, identifying receptors, enzymes and signal transducers that can be targeted easier than other molecules (**Figure 3A**). Of these druggable interfaces, 17 targets were identified as associated with GO biological function of ‘response to stress’. Two of 17 were kinases, of which only one, PRKAB1(β 1 subunit of the metabolic master regulator, AMPK) had commercially available and extensively validated specific and potent agonists with known structural basis (14-16) (**Figure 3A**). When proteins encoded by C#1-6 were analyzed for cooperativity between cellular processes within protein-protein interaction (PPI) networks using STRING (17), PRKAB1 and other subunits of AMPK appeared at the crossroads between ‘pathogen-sensing’, ‘autophagy’ and epithelial ‘tight and adherens junctions’ and ‘polarity complexes’, modules (**Figure S5**). This was hardly surprising because AMPK’s role in the stabilization of the gut barrier has been known for over a decade (9-11); however, beyond weak agonists like mesalamine (12, 13), nothing has emerged, and the use of relatively more potent but non-specific agonists such as Metformin in IBD is limited due to symptoms of GI intolerance.

Proposed mechanism of action of PRKAB1 in the gut lining

We hypothesized that our AI-guided approach may have pinpointed a specific subtype of AMPK (i.e., trimers of the kinase that includes PRKA β 1) that is important, and that PRKAB1-specific agonists may offer a higher degree of precision and efficacy over non-specific AMPK agonists. Mechanistically, they may augment epithelial tight junctions (TJs) in the presence of pathogens by activating a specialized signaling program in the epithelium lining the gut, the stress polarity signaling (SPS) pathway (18) (**Figure 3B**). The SPS pathway involves the

phosphorylation of the polarity scaffold, Girdin (GRDN) at a single site (Ser245) by AMPK, an event that appears to be both necessary and sufficient for the strengthening of epithelial junctions under bioenergetic stress. Because the SPS-pathway is triggered exclusively as a stress response, and improves modular cooperativity within the PPI network, it fulfills the criteria of “creative elements” (19); the latter are believed to be critical for the evolvability of complex systems and their pharmacological modulation is predicted to help survive unprecedented challenges/stressors. More importantly, in a related manuscript (20), we confirmed using PRKAB1-specific agonists that the SPS-pathway as a putative cell-type specific ‘companion’ biomarker for AMPK activation in the gut epithelium.”

On Page #8-9 we listed additional manuscripts widely demonstrating the efficacy of AMPK agonists (namely Metformin) in mouse models of colitis:

PRKAB1-agonists ameliorate colitis in a network-rationalized murine model

It is well known that no single mouse model recapitulates all the multifaceted complexities of IBD (27). AMPK’s role (or the role of its agonists) in protecting the gut barrier has been evaluated in several murine models of colitis, including DSS(12), TNBS(21-23), IL10-/- (24, 25) and adoptive T-cell transfer models(26).

BIBLIOGRAPHY OF CITATIONS IN THIS REBUTTAL

1. S. Singh, M. H. Murad, M. Fumery, P. S. Dulai, W. J. Sandborn, First- and Second-Line Pharmacotherapies for Patients With Moderate to Severely Active Ulcerative Colitis: An Updated Network Meta-Analysis. *Clin Gastroenterol Hepatol* **18**, 2179-2191 e2176 (2020); published online EpubSep (10.1016/j.cgh.2020.01.008).
2. L. A. Peters, J. Perrigoue, A. Mortha, A. Iuga, W. M. Song, E. M. Neiman, S. R. Llewellyn, A. Di Narzo, B. A. Kidd, S. E. Telesco, Y. Zhao, A. Stojmirovic, J. Senddecki, K. Shameer, R. Miotto, B. Losic, H. Shah, E. Lee, M. Wang, J. J. Faith, A. Kasarskis, C. Brodmerkel, M. Curran, A. Das, J. R. Friedman, Y. Fukui, M. B. Humphrey, B. M. Iritani, N. Sibinga, T. K. Tarrant, C. Argmann, K. Hao, P. Roussos, J. Zhu, B. Zhang, R. Dobrin, L. F. Mayer, E. E. Schadt, A functional genomics predictive network model identifies regulators of inflammatory bowel disease. *Nat Genet* **49**, 1437-1449 (2017); published online EpubOct (10.1038/ng.3947).
3. B. Verstockt, S. Verstockt, J. Dehairs, V. Ballet, H. Blevi, W. J. Wollants, C. Breynaert, G. Van Assche, S. Vermeire, M. Ferrante, Low TREM1 expression in whole blood predicts anti-TNF response in inflammatory bowel disease. *EBioMedicine* **40**, 733-742 (2019); published online EpubFeb (10.1016/j.ebiom.2019.01.027).
4. W. Vanhove, P. M. Peeters, D. Staelens, A. Schraenen, J. Van der Goten, I. Cleynen, S. De Schepper, L. Van Lommel, N. L. Reynaert, F. Schuit, G. Van Assche, M. Ferrante, G. De Hertogh, E. F. Wouters, P. Rutgeerts, S. Vermeire, K. Nys, I. Arijs, Strong Upregulation of AIM2 and IFI16 Inflammasomes in the Mucosa of Patients with Active Inflammatory Bowel Disease. *Inflamm Bowel Dis* **21**, 2673-2682 (2015); published online EpubNov (10.1097/MIB.0000000000000535).
5. J. Van der Goten, W. Vanhove, K. Lemaire, L. Van Lommel, K. Machiels, W. J. Wollants, V. De Preter, G. De Hertogh, M. Ferrante, G. Van Assche, P. Rutgeerts, F. Schuit, S. Vermeire, I. Arijs, Integrated miRNA and mRNA expression profiling in inflamed colon of patients with ulcerative colitis. *PLoS One* **9**, e116117 (2014)10.1371/journal.pone.0116117).

6. I. Arijs, G. De Hertogh, K. Lemaire, R. Quintens, L. Van Lommel, K. Van Steen, P. Leemans, I. Cleynen, G. Van Assche, S. Vermeire, K. Geboes, F. Schuit, P. Rutgeerts, Mucosal gene expression of antimicrobial peptides in inflammatory bowel disease before and after first infliximab treatment. *PLoS One* **4**, e7984 (2009); published online EpubNov 24 (10.1371/journal.pone.0007984).
7. I. Arijs, G. De Hertogh, B. Lemmens, L. Van Lommel, M. de Bruyn, W. Vanhove, I. Cleynen, K. Machiels, M. Ferrante, F. Schuit, G. Van Assche, P. Rutgeerts, S. Vermeire, Effect of vedolizumab (anti-alpha4beta7-integrin) therapy on histological healing and mucosal gene expression in patients with UC. *Gut* **67**, 43-52 (2018); published online EpubJan (10.1136/gutjnl-2016-312293).
8. J. Pekow, U. Dougherty, Y. Huang, E. Gometz, J. Nathanson, G. Cohen, S. Levy, M. Kocherginsky, N. Venu, M. Westerhoff, J. Hart, A. E. Noffsinger, S. B. Hanauer, R. D. Hurst, A. Fichera, L. J. Joseph, Q. Liu, M. Bissonnette, Gene signature distinguishes patients with chronic ulcerative colitis harboring remote neoplastic lesions. *Inflamm Bowel Dis* **19**, 461-470 (2013); published online EpubMar (10.1097/MIB.0b013e3182802bac).
9. X. Sun, Q. Yang, C. J. Rogers, M. Du, M. J. Zhu, AMPK improves gut epithelial differentiation and barrier function via regulating Cdx2 expression. *Cell Death Differ* **24**, 819-831 (2017); published online EpubMay (10.1038/cdd.2017.14).
10. X. Sun, M. J. Zhu, AMP-activated protein kinase: a therapeutic target in intestinal diseases. *Open Biol* **7**, (2017); published online EpubAug (10.1098/rsob.170104).
11. M. J. Zhu, X. Sun, M. Du, AMPK in regulation of apical junctions and barrier function of intestinal epithelium. *Tissue Barriers* **6**, 1-13 (2018)10.1080/21688370.2018.1487249).
12. L. Chen, J. Wang, Q. You, S. He, Q. Meng, J. Gao, X. Wu, Y. Shen, Y. Sun, X. Wu, Q. Xu, Activating AMPK to Restore Tight Junction Assembly in Intestinal Epithelium and to Attenuate Experimental Colitis by Metformin. *Front Pharmacol* **9**, 761 (2018)10.3389/fphar.2018.00761).
13. H. Park, W. Kim, D. Kim, S. Jeong, Y. Jung, Mesalazine Activates Adenosine Monophosphate-activated Protein Kinase: Implication in the Anti-inflammatory Activity of this Anti-colitic Drug. *Curr Mol Pharmacol* **12**, 272-280 (2019)10.2174/1874467212666190308103448).
14. B. Xiao, M. J. Sanders, D. Carmena, N. J. Bright, L. F. Haire, E. Underwood, B. R. Patel, R. B. Heath, P. A. Walker, S. Hallen, F. Giordanetto, S. R. Martin, D. Carling, S. J. Gamblin, Structural basis of AMPK regulation by small molecule activators. *Nat Commun* **4**, 3017 (2013)10.1038/ncomms4017).
15. C. T. Salatto, R. A. Miller, K. O. Cameron, E. Cokorinos, A. Reyes, J. Ward, M. F. Calabrese, R. G. Kurumbail, F. Rajamohan, A. S. Kalgutkar, D. A. Tess, A. Shavnya, N. E. Genung, D. J. Edmonds, A. Jatkar, B. S. Maciejewski, M. Amaro, H. Gandhok, M. Monetti, K. Cialdea, E. Bollinger, J. M. Kreeger, T. M. Coskran, A. C. Opsahl, G. G. Boucher, M. J. Birnbaum, P. DaSilva-Jardine, T. Rolph, Selective Activation of AMPK beta1-Containing Isoforms Improves Kidney Function in a Rat Model of Diabetic Nephropathy. *J Pharmacol Exp Ther* **361**, 303-311 (2017); published online EpubMay (10.1124/jpet.116.237925).
16. K. O. Cameron, D. W. Kung, A. S. Kalgutkar, R. G. Kurumbail, R. Miller, C. T. Salatto, J. Ward, J. M. Withka, S. K. Bhattacharya, M. Boehm, K. A. Borzilleri, J. A. Brown, M. Calabrese, N. L. Caspers, E. Cokorinos, E. L. Conn, M. S. Dowling, D. J. Edmonds, H. Eng, D. P. Fernando, R. Frisbie, D. Hepworth, J. Landro, Y. Mao, F. Rajamohan, A. R. Reyes, C. R. Rose, T. Ryder, A. Shavnya, A. C. Smith, M. Tu, A. C. Wolford, J. Xiao, Discovery and Preclinical Characterization of 6-Chloro-5-[4-(1-hydroxycyclobutyl)phenyl]-1H-indole-3-carboxylic Acid (PF-06409577), a Direct Activator of Adenosine Monophosphate-activated Protein Kinase (AMPK), for the Potential Treatment of Diabetic Nephropathy. *J Med Chem* **59**, 8068-8081 (2016); published online EpubSep 8 (10.1021/acs.jmedchem.6b00866).
17. D. Szklarczyk, J. H. Morris, H. Cook, M. Kuhn, S. Wyder, M. Simonovic, A. Santos, N. T. Doncheva, A. Roth, P. Bork, L. J. Jensen, C. von Mering, The STRING database in 2017: quality-controlled protein-protein association networks, made broadly accessible. *Nucleic Acids Res* **45**, D362-D368 (2017); published online EpubJan 4 (10.1093/nar/gkw937).

18. N. Aznar, A. Patel, C. C. Rohena, Y. Dunkel, L. P. Joosen, V. Taupin, I. Kufareva, M. G. Farquhar, P. Ghosh, AMP-activated protein kinase fortifies epithelial tight junctions during energetic stress via its effector GIV/Girdin. *Elife* **5**, (2016); published online EpubNov 04 (10.7554/eLife.20795).
19. P. Csermely, Creative elements: network-based predictions of active centres in proteins and cellular and social networks. *Trends Biochem Sci* **33**, 569-576 (2008); published online EpubDec (10.1016/j.tibs.2008.09.006).
20. P. Ghosh, L. Swanson, I. M. Sayed, Y. Mittal, B. B. Lim, S. R. Ibeawuchi, M. Foretz, B. Viollet, D. Sahoo, S. Das, The stress polarity signaling (SPS) pathway serves as a marker and a target in the leaky gut barrier: implications in aging and cancer. *Life Sci Alliance* **3**, (2020); published online EpubMar (10.26508/lsa.201900481).
21. A. Bai, A. G. Ma, M. Yong, C. R. Weiss, Y. Ma, Q. Guan, C. N. Bernstein, Z. Peng, AMPK agonist downregulates innate and adaptive immune responses in TNBS-induced murine acute and relapsing colitis. *Biochem Pharmacol* **80**, 1708-1717 (2010); published online EpubDec 1 (10.1016/j.bcp.2010.08.009).
22. M. Takahara, A. Takaki, S. Hiraoka, T. Adachi, Y. Shimomura, H. Matsushita, T. T. T. Nguyen, K. Koike, A. Ikeda, S. Takashima, Y. Yamasaki, T. Inokuchi, H. Kinugasa, Y. Sugihara, K. Harada, S. Eikawa, H. Morita, H. Udono, H. Okada, Berberine improved experimental chronic colitis by regulating interferon-gamma- and IL-17A-producing lamina propria CD4(+) T cells through AMPK activation. *Sci Rep* **9**, 11934 (2019); published online EpubAug 15 (10.1038/s41598-019-48331-w).
23. B. Xu, Y. L. Li, M. Xu, C. C. Yu, M. Q. Lian, Z. Y. Tang, C. X. Li, Y. Lin, Geniposide ameliorates TNBS-induced experimental colitis in rats via reducing inflammatory cytokine release and restoring impaired intestinal barrier function. *Acta Pharmacol Sin* **38**, 688-698 (2017); published online EpubMay (10.1038/aps.2016.168).
24. S. J. Koh, J. M. Kim, I. K. Kim, S. H. Ko, J. S. Kim, Anti-inflammatory mechanism of metformin and its effects in intestinal inflammation and colitis-associated colon cancer. *J Gastroenterol Hepatol* **29**, 502-510 (2014); published online EpubMar (10.1111/jgh.12435).
25. Y. Xue, H. Zhang, X. Sun, M. J. Zhu, Metformin Improves Ileal Epithelial Barrier Function in Interleukin-10 Deficient Mice. *PLoS One* **11**, e0168670 (2016)10.1371/journal.pone.0168670).
26. J. Blagih, F. Coulombe, E. E. Vincent, F. Dupuy, G. Galicia-Vazquez, E. Yurchenko, T. C. Raissi, G. J. van der Windt, B. Viollet, E. L. Pearce, J. Pelletier, C. A. Piccirillo, C. M. Krawczyk, M. Divangahi, R. G. Jones, The energy sensor AMPK regulates T cell metabolic adaptation and effector responses in vivo. *Immunity* **42**, 41-54 (2015); published online EpubJan 20 (10.1016/j.immuni.2014.12.030).
27. J. A. Jiminez, T. C. Uwiera, G. Douglas Inglis, R. R. Uwiera, Animal models to study acute and chronic intestinal inflammation in mammals. *Gut Pathog* **7**, 29 (2015)10.1186/s13099-015-0076-y).

Reviewers' Comments:

Reviewer #1:

Remarks to the Author:

In the revision of their manuscript, "AI-guided discovery of a barrier-protective therapy in inflammatory bowel disease", Sahoo et al. have gone to significant lengths to address all of the reviewer concerns and have significantly revised the manuscript as a result, including expansion of the methods and supplementary material to enable a better understanding of all that was done. I think at this point the level of detail is reasonably adequate and areas where claims were stronger than what was supported by the results, have been more appropriately caveated or weakened. I think the approach taken will be of broad interest not just to the IBD community, but to a broader readership interested in how high dimensional data can be more appropriately organized and processed to generate informative hypotheses or support existing hypotheses.

I am generally good with the revision and think the results are well enough supported by the additional analyses and experimentation. But there are still parts of the analyses that are not well described, where given the claims, there is a want to understand to the point of being able to have confidence in the claims made. The one that stands more out to me, given it involves methodologies I am pretty familiar with, relates to the comparison of methods to distinguish responders from non-responders, with the claim that the author's approach does better than other approaches such as the differential and Bayesian approaches. What tuned me into this issue was a statement in the main paragraph of the brief Methods section, "This offers a distinct advantage from conventional computational methods (Bayesian, Differential, etc.) that rely exclusively on symmetric linear relationships in networks." First only Bayesian and Differential are highlighted in the comparison so not sure what the "etc." refers to, and second I do not believe the statement is correct in that Bayesian networks constructed from discretized data are capable of capturing nonlinear relationships and through the use of structure priors can also introduce a significant degree of asymmetry. Thus it is just not clear to me what the authors are claiming.

This is a relatively minor point since it really isn't the aim of the paper to do an in-depth methodological comparison, but in this case it is impossible to know what the differences indicated by the authors in figure 5 regarding responder/non-responder prediction, actually mean. Since how the Bayesian and differential approaches were applied are still not discussed, other than the authors seeming to indicate they ran the methods more or less out of the box (indicated they ran them according to instructions), but these methods, the Bayesian network approach in particular, are not really out of the box solutions, but rather have a wide array of parameters that can be set and tuned, different types of priors that can be employed and so on, and so I would claim that if an more out of the box approach was taken, that it's not really a fair comparison, or at least it is impossible to assess it without knowing how the data were normalized for input into the program, how many iterations were run, how resulting networks were derived, how the predictions off of these networks were made in terms of responder/non-responder and so on. And honestly I am not sure why there is the want to push these comparisons in this way given it is more a distraction and just has me wanting to really understand whether the difference indicated is real or just an artifact of how the analyses were run.

Reviewer #2:

Remarks to the Author:

The authors have addressed all my concerns.

Check sentence on page 6 (207-209), I believe is lacking a verb.

Reviewer #4:

Remarks to the Author:

In their manuscript entitled "AI-guided discovery of a barrier protective therapy in Inflammatory bowel disease", the authors propose a new pipeline for drug discovery and applied this concept to Inflammatory Bowel Disease.

The novelty is related to a Boolean analysis of the biological network in IBD. 1) This approach allowed the authors to define and organize the key biological functions involved in disease mechanisms. 2) Based on these results, they explore the best drugable genes and focused on an AMPK subunit expressed in the intestine. Experiments performed with animal models and organoids derived from healthy and diseased people further supported their proposition. 3) Finally, the authors explored the robustness of their discovery pipeline using the results of clinical trials performed in IBD patients.

As a whole, this is an excellent work and I have no concern on the revised version of the paper.
Jean-Pierre Hugot

REVIEWER COMMENTS

Reviewer #1 (Remarks to the Author):

In the revision of their manuscript, "AI-guided discovery of a barrier-protective therapy in inflammatory bowel disease", Sahoo et al. have gone to significant lengths to address all of the reviewer concerns and have significantly revised the manuscript as a result, including expansion of the methods and supplementary material to enable a better understanding of all that was done. I think at this point the level of detail is reasonably adequate and areas where claims were stronger than what was supported by the results, have been more appropriately caveated or weakened. I think the approach taken will be of broad interest not just to the IBD community, but to a broader readership interested in how high dimensional data can be more appropriately organized and processed to generate informative hypotheses or support existing hypotheses.

I am generally good with the revision and think the results are well enough supported by the additional analyses and experimentation. But there are still parts of the analyses that are not well described, where given the claims, there is a want to understand to the point of being able to have confidence in the claims made. The one that stands more out to me, given it involves methodologies I am pretty familiar with, relates to the comparison of methods to distinguish responders from non-responders, with the claim that the author's approach does better than other approaches such as the differential and Bayesian approaches. What tuned me into this issue was a statement in the main paragraph of the brief Methods section, "This offers a distinct advantage from conventional computational methods (Bayesian, Differential, etc.) that rely exclusively on symmetric linear relationships in networks." First only Bayesian and Differential are highlighted in the comparison so not sure what the "etc." refers to, and second I do not believe the statement is correct in that Bayesian networks constructed from discretized data are capable of capturing nonlinear relationships and through the use of structure priors can also introduce a significant degree of asymmetry. Thus it is just not clear to me what the authors are claiming.

This is a relatively minor point since it really isn't the aim of the paper to do an in-depth methodological comparison, but in this case it is impossible to know what the differences indicated by the authors in figure 5 regarding responder/non-responder prediction, actually mean. Since how the Bayesian and differential approaches were applied are still not discussed, other than the authors seeming to indicate they ran the methods more or less out of the box (indicated they ran them according to instructions), but these methods, the Bayesian network approach in particular, are not really out of the box solutions, but rather have a wide array of parameters that can be set and tuned, different types of priors that can be employed and so on, and so I would claim that if an more out of the box approach was taken, that it's not really a fair comparison, or at least it is impossible to assess it without knowing how the data were normalized for input into the program, how many iterations were run, how resulting networks were derived, how the predictions off of these networks were made in terms of responder/non-responder and so on. And honestly I am not sure why there is the want to push these comparisons in this way given it is more a distraction and just has me wanting to really understand whether the difference indicated is real or just an artifact of how the analyses were run.

Reviewer #2 (Remarks to the Author):

The authors have addressed all my concerns.

Check sentence on page 6 (207-209), I believe is lacking a verb.

Reviewer #4 (Remarks to the Author):

In their manuscript entitled "AI-guided discovery of a barrier protective therapy in Inflammatory bowel disease", the authors propose a new pipeline for drug discovery and applied this concept to Inflammatory Bowel Disease.

The novelty is related to a Boolean analysis of the biological network in IBD. 1) This approach allowed the authors to define and organize the key biological functions involved in disease mechanisms. 2) Based on these results, they explore the best drugable genes and focused on an AMPK subunit expressed in the intestine. Experiments performed with animal models and organoids derived from healthy and diseased people further supported their proposition. 3) Finally, the authors explored the robustness of their discovery pipeline using the results of clinical trials performed in IBD patients.

As a whole, this is an excellent work and I have no concern on the revised version of the paper.

Jean-Pierre Hugot

POINT BY POINT RESPONSE TO REVIEWER'S CRITIQUES

Reviewer #1 (Remarks to the Author):

In the revision of their manuscript, "AI-guided discovery of a barrier-protective therapy in inflammatory bowel disease", Sahoo et al. have gone to significant lengths to address all of the reviewer concerns and have significantly revised the manuscript as a result, including expansion of the methods and supplementary material to enable a better understanding of all that was done. I think at this point the level of detail is reasonably adequate and areas where claims were stronger than what was supported by the results, have been more appropriately caveated or weakened. I think the approach taken will be of broad interest not just to the IBD community, but to a broader readership interested in how high dimensional data can be more appropriately organized and processed to generate informative hypotheses or support existing hypotheses.

Response: We are grateful to the reviewer #1 for these comments, and that he/she appreciated the diligence that we demonstrated in being responsive to the critiques during round #1 of revisions. It is particularly encouraging and gratifying for us to see that he/she sees the scope and power of our approach and its potential to impact drug discovery (and hence of interest) in fields beyond IBD.

I am generally good with the revision and think the results are well enough supported by the additional analyses and experimentation. But there are still parts of the analyses that are not well described, where given the claims, there is a want to understand to the point of being able to have confidence in the claims made. The one that stands more out to me, given it involves methodologies I am pretty familiar with, relates to the comparison of methods to distinguish responders from non-responders, with the claim that the author's approach does better than other approaches such as the differential and Bayesian approaches. What tuned me into this issue was a statement in the main paragraph of the brief Methods section, "This offers a distinct advantage from conventional computational methods (Bayesian, Differential, etc.) that rely exclusively on symmetric linear relationships in networks." First only Bayesian and Differential are highlighted in the comparison so not sure what the "etc." refers to, and second I do not believe the statement is correct in that Bayesian networks constructed from discretized data are capable of capturing nonlinear relationships and through the use of structure priors can also introduce a significant degree of asymmetry. Thus it is just not clear to me what the authors are claiming.

Response: We agree with the reviewer that in general Bayesian analysis can capture nonlinear relationship using priors. For example, in the Bayesian analysis framework described by Peters et al. Nat Gen 2017, a gene co-expression network is built using the gene expression data and asymmetry is introduced using independent eQTL data (priors). We refer to this Bayesian analysis framework that uses symmetric linear relationships from gene expression data. However, our approach extracts asymmetric relationships directly from the same dataset without any additional information (such as priors). Mathematically, it is also possible to extract similar relationships using Bayesian analysis. We have revised the sentence (main text, methods) to

clarify our viewpoint. “etc.” is referring to other network-based approaches such as correlation, co-expression, and mutual information network. For the convenience of the reviewer, we have copied and pasted that section below:

“This offers a distinct advantage from currently used conventional computational methods that rely exclusively on symmetric linear relationships from gene expression data, e.g. Differential, correlation-network, coexpression-network, mutual information-network, and the Bayesian signature that was originally identified using one of the test cohorts, GSE83687 (17).”

This is a relatively minor point since it really isn't the aim of the paper to do an in-depth methodological comparison, but in this case it is impossible to know what the differences indicated by the authors in figure 5 regarding responder/non-responder prediction, actually mean. Since how the Bayesian and differential approaches were applied are still not discussed, other than the authors seeming to indicate they ran the methods more or less out of the box (indicated they ran them according to instructions), but these methods, the Bayesian network approach in particular, are not really out of the box solutions, but rather have a wide array of parameters that can be set and tuned, different types of priors that can be employed and so on, and so I would claim that if a more out of the box approach was taken, that it's not really a fair comparison, or at least it is impossible to assess it without knowing how the data were normalized for input into the program, how many iterations were run, how resulting networks were derived, how the predictions off of these networks were made in terms of responder/non-responder and so on. And honestly I am not sure why there is the want to push these comparisons in this way given it is more a distraction and just has me wanting to really understand whether the difference indicated is real or just an artifact of how the analyses were run.

Response: We have directly used the Bayesian analysis results from Peters et al. 2017 (GSE83687) for comparison. For Differential analysis, we used Peters dataset and performed standard t-tests adjusted at 1% FDR using the Benjamini-Hochberg Procedure. We have added a new section “**Comparison of Boolean with Bayesian and Differential analyses**” in the supplementary methods.

Reviewer #2 (Remarks to the Author):

The authors have addressed all my concerns.

Check sentence on page 6 (207-209), I believe is lacking a verb.

Response to reviewer #2: We have now re-checked the sentence on Page 6. The missing verb “serves” is now added and the sentence reads as: *“More importantly, we recently confirmed using PRKAB1-specific agonists that the SPS-pathway serves as a putative cell-type specific ‘companion’ biomarker for AMPK activation in the gut epithelium (48).”*

Reviewer #4 (Remarks to the Author):

In their manuscript entitled “AI-guided discovery of a barrier protective therapy in Inflammatory bowel disease”, the authors propose a new pipeline for drug discovery and applied this concept to Inflammatory Bowel Disease.

The novelty is related to a Boolean analysis of the biological network in IBD. 1) This approach allowed the authors to define and organize the key biological functions involved in disease mechanisms. 2) Based on these results, they explore the best drugable genes and focused on an AMPK subunit expressed in the intestine. Experiments performed with animal models and organoids derived from healthy and diseased people further supported their proposition. 3) Finally, the authors explored the robustness of their discovery pipeline using the results of clinical trials performed in IBD patients.

As a whole, this is an excellent work and I have no concern on the revised version of the paper.

Response to reviewer #4: We are grateful to this reviewer for his/her generous complements regarding the novelty, quality, and scope of our work.

Reviewers' Comments:

Reviewer #1:

Remarks to the Author:

The authors have adequately addressed my concerns and those of the other reviewers.